# A conserved dimer interface connects ERH and YTH family proteins to promote gene silencing

Guodong Xie[1], Tommy V. Vo[2], Gobi Thillainadesan[2], Sahana Holla[2], Beibei Zhang[1], Yiyang Jiang[1], Mengqi Lv[1], Zheng Xu[1], Chongyuan Wang[1], Vanivilasini Balachandran[2], Yunyu Shi[1], Fudong Li[1] & Shiv I.S. Grewal [ID] [2]

Gene regulatory mechanisms rely on a complex network of RNA processing factors to prevent untimely gene expression. In fission yeast, the highly conserved ortholog of human ERH, called Erh1, interacts with the YTH family RNA binding protein Mmi1 to form the Erh1-Mmi1 complex (EMC) implicated in gametogenic gene silencing. However, the structural basis of EMC assembly and its functions are poorly understood. Here, we present the co-crystal structure of the EMC that consists of Erh1 homodimers interacting with Mmi1 in a 2:2 stoichiometry via a conserved molecular interface. Structure-guided mutation of the Mmi1[Trp112] residue, which is required for Erh1 binding, causes defects in facultative heterochromatin assembly and gene silencing while leaving Mmi1-mediated transcription termination intact. Indeed, EMC targets masked in *mmi1Δ* due to termination defects are revealed in *mmi1[W112A]*. Our study delineates EMC requirements in gene silencing and identifies an ERH interface required for interaction with an RNA binding protein.

---

[1] Hefei National Laboratory for Physical Sciences at the Microscale, School of Life Sciences, University of Science and Technology of China, 230026 Hefei, China. [2] Laboratory of Biochemistry and Molecular Biology, National Cancer Institute, National Institutes of Health, Bethesda, MD 20892, USA. These authors contributed equally: Guodong Xie, Tommy V. Vo. Correspondence and requests for materials should be addressed to F.L. (email: lifudong@ustc.edu.cn) or to S.I.S.G. (email: grewals@mail.nih.gov)

Gene expression is tightly regulated to ensure accurate translation of genetic information in response to developmental and environmental signals. A variety of gene regulatory mechanisms generate diverse expression patterns reflecting the cell type or specific developmental stage[1,2]. In addition to transcriptional gene control[3–5], factors acting at the level of post-transcriptional processing have emerged as critical players in defining gene expression profiles and preventing ectopic expression of genetic information[6–9]. Indeed, untimely gene expression is a major cause of diseases including cancer[10,11]. Despite major advances in our understanding of conserved gene regulatory pathways, mechanisms underlying developmental or environmental control of gene expression remain poorly understood

The fission yeast *Schizosaccharomyces pombe* contains highly conserved RNA processing and chromatin-modifying activities, thus providing an excellent model system for exploring gene regulatory mechanisms[4]. One such regulatory mechanism controls major developmental changes that occur in response to nutrient starvation. In cells starved of nitrogen, the switch from the mitotic to the meiotic cell cycle requires the coordinated activation of hundreds of gametogenic genes involved in meiosis and sexual differentiation[12]. During the mitotic cell cycle, silencing of these genes[9] requires a highly conserved protein named *Enhancer of rudimentary homolog 1* (Erh1)[13–15] belonging to the ERH protein family implicated in various nuclear processes[16]. Erh1 associates with nuclear RNA elimination factors, including MTREC (PAXT in mammals[17]), which is composed of the zinc-finger protein, Red1, and the Mtr4-like protein, Mtl1, as well as the CCR4-NOT complex that act together with other factors to facilitate RNA degradation by the 3′→5′ exonuclease Rrp6 and RNAi machinery[15,18–25]. Moreover, Erh1 and its interaction partners have been shown to mediate targeting of the histone 3 lysine-9 (H3K9) methyltransferase Clr4 (a homolog of mammalian Suv39h) to assemble facultative heterochromatin at meiotic genes[26–28]. This provides an additional level of gene silencing that offers protection from the deleterious effects of improper meiotic gene expression during the mitotic cell cycle.

Many gametogenic gene transcripts silenced by Erh1 contain a determinant of selective removal (DSR) element that is recognized by the YTH-domain of the RNA binding protein Mmi1[18,29–33]. Recent work has revealed that Mmi1 interacts with Erh1 to form a complex, called EMC[14], and loss of Mmi1 affects recruitment of Erh1 and its associated RNA processing activities to target transcripts[14,27]. Despite these advances, the structural features of EMC and the functional significance of complex formation for silencing gametogenic genes and assembling facultative heterochromatin has remained enigmatic. Moreover, it remains unclear whether Mmi1 association with Erh1 is critical for the diverse functions attributed to Mmi1, including its recently described role in non-canonical transcription termination of meiotic mRNAs and regulatory long non-coding RNAs (lncRNAs)[34–36].

Here, we present the co-crystal structure of Erh1 in complex with the amino-terminal domain of Mmi1. Our structure reveals that Mmi1 binds homodimers of Erh1 in a 2:2 stoichiometry, via a conserved molecular interface characteristic of ERH family proteins. Structure-guided mutational analysis shows that the Mmi1-Erh1 interaction is essential for facultative heterochromatin assembly and for silencing of gametogenic genes. Interestingly, we discover that an important function of Mmi1 in promoting non-canonical transcription termination at meiotic genes, and in preventing lncRNAs from invading and repressing adjacent genes, is not dependent upon its association with Erh1. Therefore, we not only reveal a distinct requirement for EMC among the various functional roles attributed to Mmi1 but also discover that the structural basis for EMC assembly involves a highly conserved ERH dimer interface.

## Results

**Erh1 interacts with the amino-terminal domain of Mmi1.** Erh1 and Mmi1 form a protein complex called EMC[14], but the nature of the interaction between these factors had not yet been characterized. To study this, we set out to determine the precise domain in Mmi1 that binds to Erh1. Mmi1 is a 488 amino acid protein that consists of a carboxy-terminal YTH domain and a largely unstructured amino-terminus (Fig. 1a). The carboxy-terminal YTH domain is known to bind to DSR motif-containing RNA[18,29–33]. However, we detected no interaction between the purified GST-tagged YTH domain of Mmi1 and His-tagged Erh1 protein (Fig. 1b).

We next tested the amino-terminal portion of Mmi1 for its ability to bind to Erh1. We were unable to obtain soluble fusion proteins encoding the GST-tagged amino-terminal (residues 1–321) region of Mmi1. Upon closer inspection, we noticed that Mmi1 residues 1–122 are highly conserved, especially within two short segments (residues 64–75 and 95–122) (Fig. 1a and Supplementary Fig. 1). We found that the purified GST-tagged Mmi1$^{1-122}$ amino-terminal fragment bound efficiently to Erh1 (Fig. 1b, c), and that an even shorter fragment containing Mmi1 residues 95–122 was sufficient for interaction (Fig. 1c, d). To confirm this finding, we performed isothermal titration calorimetry (ITC). Remarkably, Mmi1$^{95-122}$ bound to Erh1 with a dissociation constant ($K_D$) of $0.83 \pm 0.19$ μM and an $N$ value of ~1 (0.90) (Fig. 1e and Supplementary Table 1). Moreover, the further truncated Mmi1$^{106-122}$ fragment showed decreased binding to Erh1, while the Mmi1$^{95-111}$ peptide did not bind at all (Fig. 1d, f). Therefore, we conclude that Mmi1$^{95-122}$ includes the Erh1-interacting domain (EID) that is necessary and sufficient to mediate EMC formation.

**EMC consists of homodimers of Erh1 bound to Mmi1.** Next, we assessed the stoichiometry of Erh1 and Erh1-Mmi1$^{95-122}$ complexes by analytical size-exclusion chromatography (SEC) using a calibrated Superdex75 10/300 GL column (GE Healthcare). Erh1 eluted as a single peak at 11.41 mL, corresponding to an apparent molecular weight (MW) of 26 kDa. With a predicted molecular weight (MW) of ~13.2 kDa, this elution profile likely represents Erh1 homodimers (Fig. 1g). This is consistent with previous reports showing that human ERH protein exists as a dimer in solution[37].

We next purified SUMO-tagged Mmi1$^{95-122}$ and incubated it with Erh1, followed by removal of the SUMO tag by ULP1 enzyme. The association of Erh1 with Mmi1$^{95-122}$ resulted in a shift of about 0.25 mL in elution volume to ~11.16 mL, with a calculated MW of 33 kDa. This result is in accordance with the expected MW of two Erh1 molecules forming a complex with two Mmi1$^{95-122}$ peptides (Fig. 1g), which was further confirmed by the $N$ value of the ITC result (Fig. 1e). Together, these results suggest that binding of the Mmi1$^{95-122}$ domain to Erh1 does not disrupt the Erh1 homodimer interface and that EMC forms in a 2:2 Erh1-Mmi1 stoichiometry.

**Co-crystal structure of EMC.** We determined the structure of EMC by crystallizing Erh1 in complex with Mmi1$^{95-122}$. We used a (Gly-Ser-Ser)$_5$ linker to fuse Erh1 and Mmi1$^{95-122}$ in tandem. The co-crystal structure was resolved to 2.7 Å resolution by molecular replacement. We refined the structure to an $R_{work}$ of 20.67%, an $R_{free}$ of 24.09% and satisfactory stereochemistry (Supplementary Table 2). Consistent with the 2:2 stoichiometry model based on the results from the SEC experiments, the co-crystal structure revealed that Erh1 forms a homodimer, with two Mmi1 subunits bound on the surface of the Erh1 homodimer (Fig. 2a, b). The final model includes residues 6–45 and 55–101 of

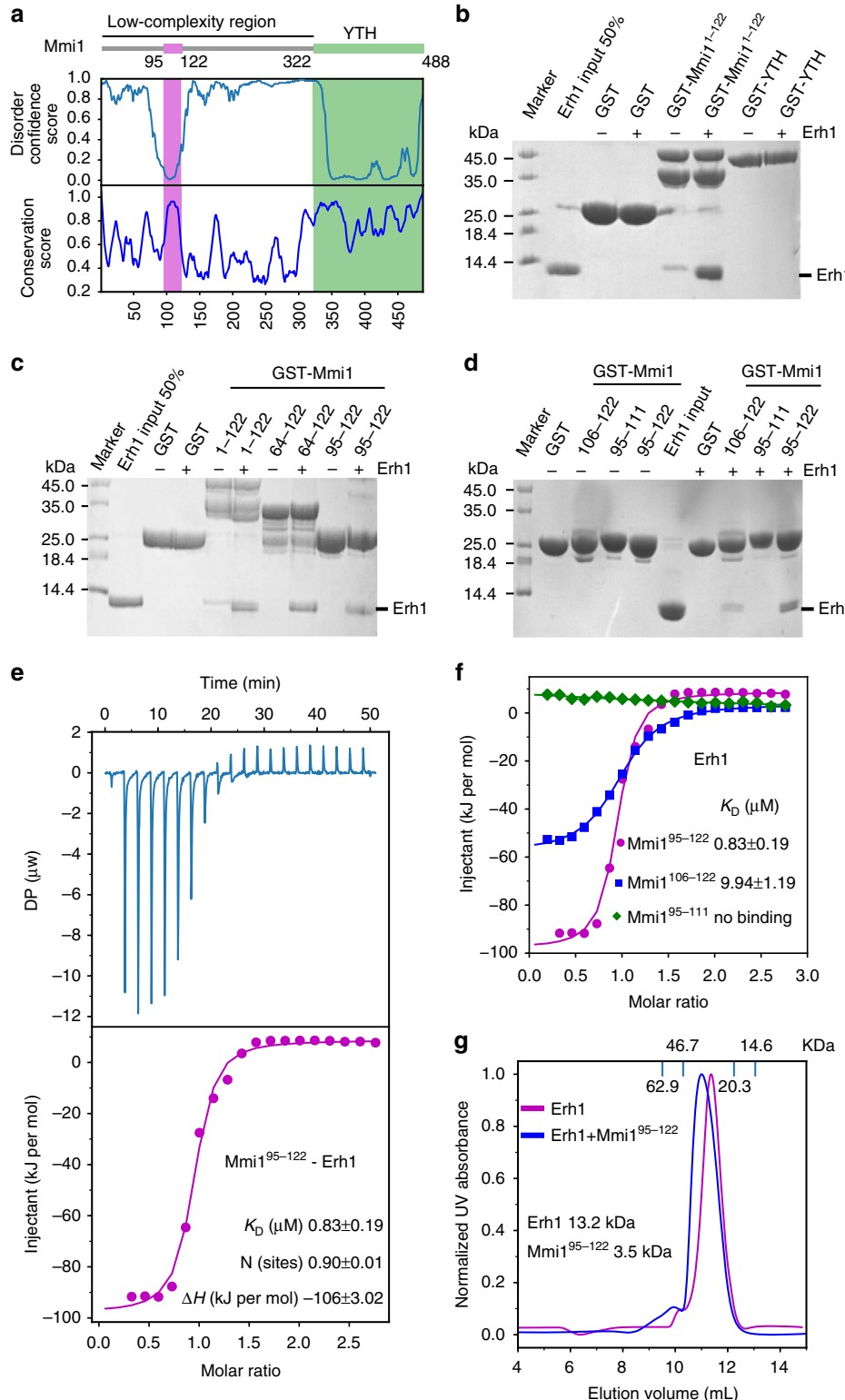

Erh1 and residues 96–119 of Mmi1 without any linker sequence (the 2Fo-Fc electron density map for residues 96–119 of Mmi1 is shown in Supplementary Fig. 2).

Overall, the Erh1 monomer adopted a typical ERH fold, characterized by a four-stranded antiparallel β-sheet (β1–β4) and three α-helixes (α1–α3), with the α-helixes on one face of the sheet (Fig. 2a). The outside faces of the β-sheets of two monomers constituted the homodimer interface and formed a pseudo-β-barrel, burying about 1000 Å² of solvent-exposed area (Fig. 2a).

At the dimer interface, hydrophobic interactions formed among the side chains of Erh1 residues Ile11, Leu13, Trp25, Leu72, Tyr81, and Pro83; two main-chain hydrogen bonds between the carbonyl oxygen and amide nitrogen atoms of Tyr81 on strand 4 of each monomer; side-chain hydrogen bonds between Arg23 and Asp27 of each monomer (Fig. 2c). Interestingly, the residues of the dimer interface are significantly conserved among divergent eukaryotic species (Fig. 3a), suggesting that dimer formation is essential for Erh1 function.

**Fig. 1** Mmi1 interacts with Erh1 in a 2:2 stoichiometry. **a** Shown is a schematic representation of the domain architecture of Mmi1. The disorder score from DISOPRED3 Server[62] and the conservation score are shown below. The conservation score is generated by a Protein Residue Conservation Prediction Server[63] from the sequence alignment of Mmi1 orthologues from *S. pombe*, *S. cryophilus*, *S. octosporus*, and *S. japonicus*. **b**–**d** Interactions of GST tagged Mmi1 peptides (GST-Mmi1) with Erh1 visualized by Coomassie blue staining. The indicated GST-Mmi1 fusion proteins or GST alone were incubated with Erh1. The complexes were collected with glutathione-agarose resin and bound proteins were eluted and then subjected to SDS-PAGE. GST or GST-Mmi1 fusion proteins without Erh1 are shown as a negative control. Source data are provided as a Source Data file. **e** The raw ITC titration data of Erh1 with SUMO tagged Mmi1$^{95-122}$ and its fitting curve are shown. $K_D$ dissociation constant, *DP* differential power, *N* binding stoichiometry, $\Delta H$ binding enthalpy. **f** ITC fitting curves of Erh1 using SUMO tagged Mmi1$^{95-122}$ (magenta), Mmi1$^{106-122}$ (blue) and Mmi1$^{95-111}$ (green) are shown. **g** Size-exclusion analysis of Erh1 (magenta) and Erh1-Mmi1$^{95-122}$ (blue). Marker sizes are 62.9 KDa (Albumin), 46.7 KDa (Ovalbumin), 20.3 KDa (Chymotrypsinogen), 14.6 KDa (Ribonuclease A). Also shown are theoretical monomer sizes (kDa) of Erh1 protein and Mmi1$^{95-122}$ peptide

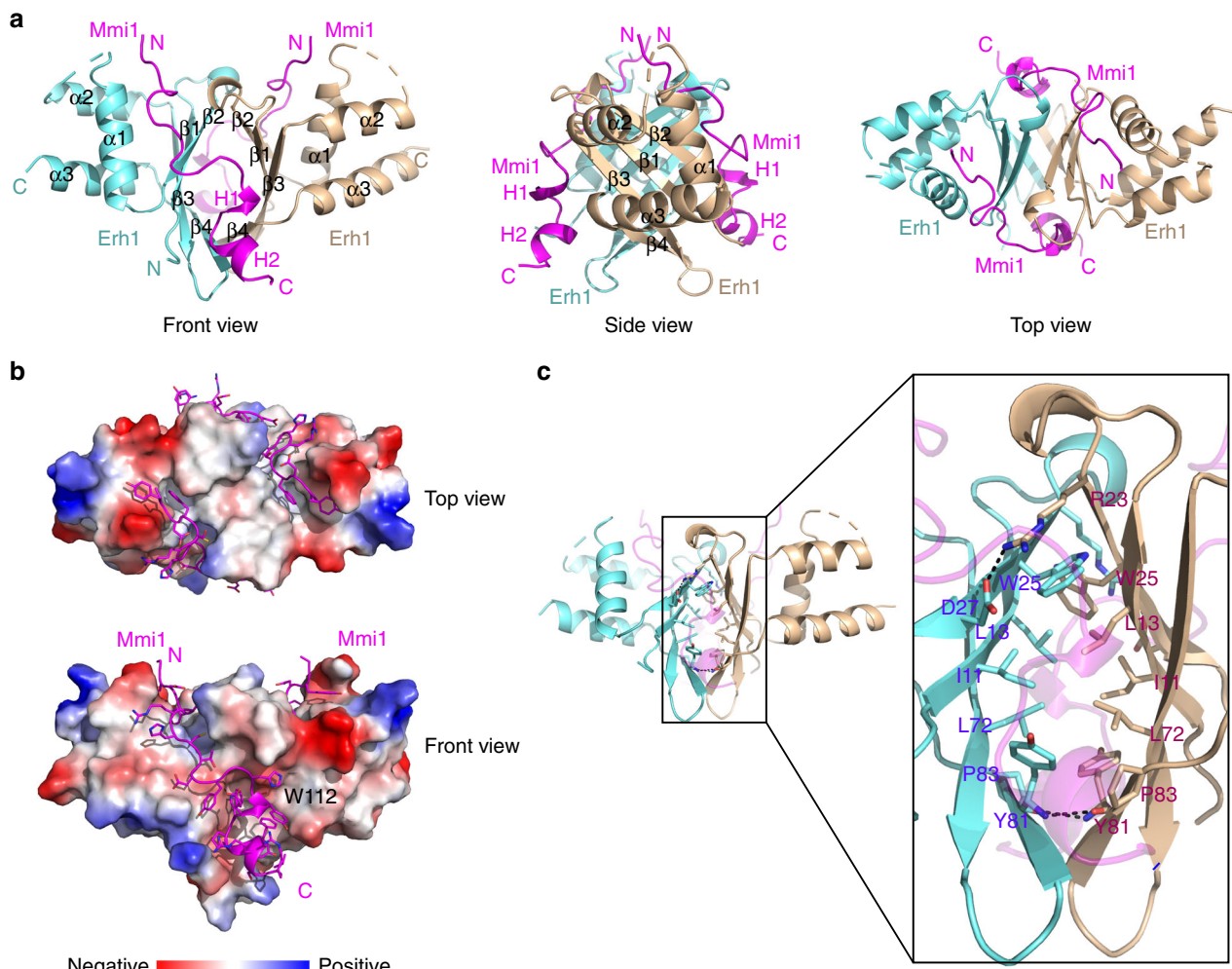

**Fig. 2** Structure of Erh1 in complex with Mmi1$^{95-122}$ peptide. **a** Ribbon representations of Erh1 bound to Mmi1$^{95-122}$ peptide. One monomer of the Erh1 homodimer is colored in aquamarine, while the other is colored in wheat. The two bound Mmi1$^{95-122}$ peptides are colored in magenta. **b** The two Mmi1$^{95-122}$ peptides are represented as sticks on the molecular face of the Erh1 homodimer. Red and blue colors denote negative and positive surface charge, respectively. **c** Ribbon representation of the interface of the Erh1 dimer. The residues at the dimer interface are shown as sticks. The hydrogen bonds formed at the dimer interface are depicted as black dashed lines

Consistent with our biochemical analyses, the structure showed two Mmi1$^{95-122}$ peptides bound on the surface of the Erh1 homodimer in a similar extended conformation, burying more than 2500 Å$^2$ of the solvent-exposed area of Erh1 (Fig. 2a, b). Each Mmi1 peptide was bound across the β2 edge of the β-sheet of each Erh1 monomer, with the amino-terminal residues (Mmi1$^{Lys96}$-Mmi1$^{Cys103}$) located in the groove between α-helix α1 and the β-sheet, whereas the carboxy-terminal residues (Mmi1$^{Thr104}$-Mmi1$^{Arg119}$) were bound to the hydrophobic core of the dimer pseudo-β-barrel (Fig. 2a). Taken together, the co-crystal structure of Erh1-Mmi1$^{95-122}$ revealed that binding of

Mmi1 to Erh1 occurs as a heterotetramer and involves an extensive set of inter- and intra-molecular interactions.

**Intra- and inter-molecular interactions stabilize EMC.** When bound to Erh1, the carboxy-terminal residues of the Mmi1$^{95-122}$ peptide ($^{109}$SYEWPYFRSLR$^{119}$) folded into two consecutive helices, consisting of a 3$_{10}$ helix (H1) and an α-helix (H2) (Figs. 2a, 3b). The two helices were connected via the Mmi1$^{Trp112}$ residue. The Mmi1$^{Trp112}$ aromatic ring stretched into a deep hydrophobic pocket formed by Erh1$^{His9}$, Erh1$^{Val70}$, Erh1$^{Ile11}$,

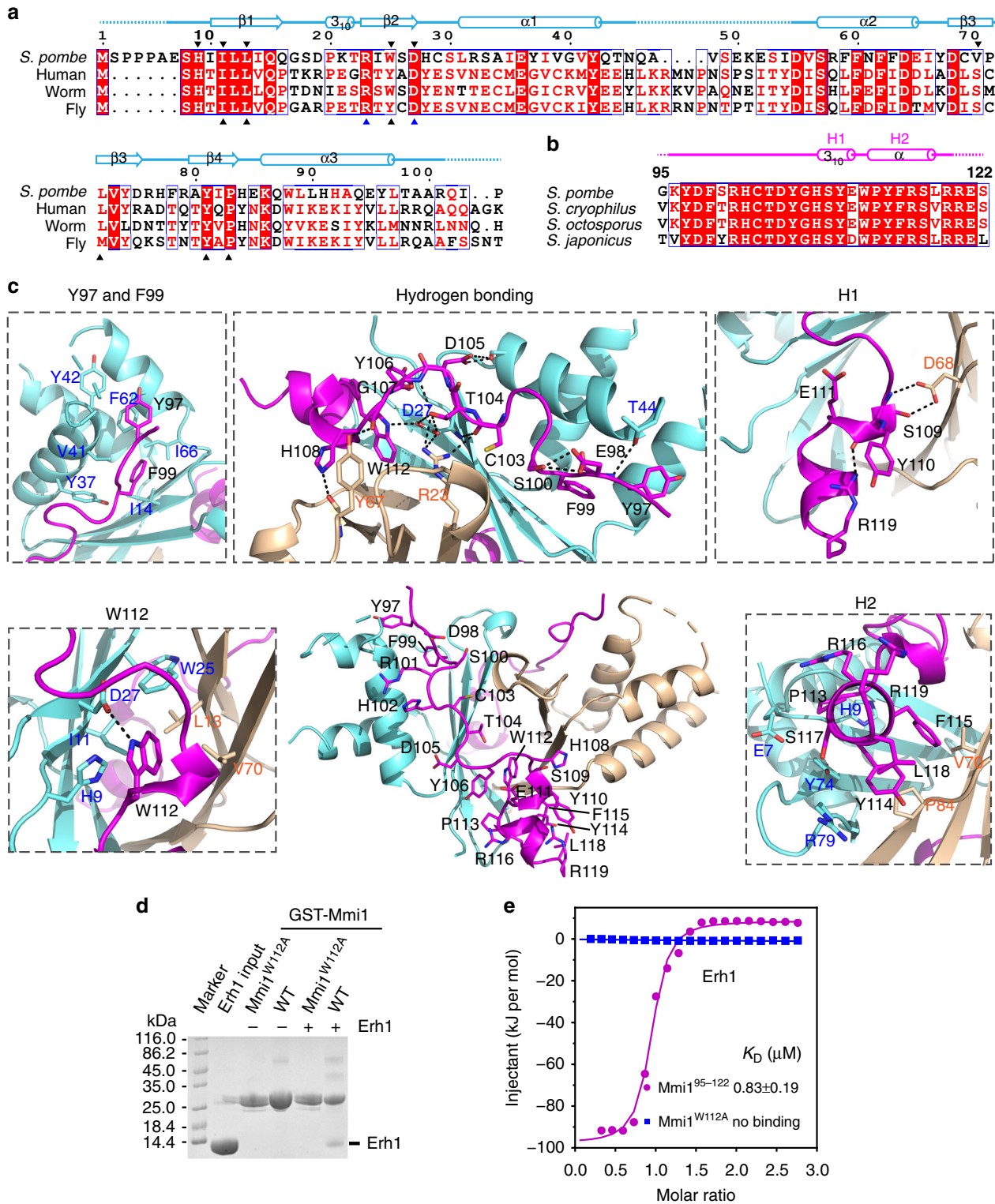

Erh1$^{Leu13}$, and Erh1$^{Trp25}$, with its Nε nitrogen forming a hydrogen bond with the side chain of Erh1$^{Asp27}$ (Fig. 3c). Indeed, Mmi1$^{Trp112}$ is crucial for EMC formation as substitution with alanine disrupted Mmi1-Erh1 interaction as confirmed by GST pull-down assay and ITC (Fig. 3d, e). Besides Mmi1$^{Trp112}$, residues Mmi1$^{Pro113}$-Mmi1$^{Arg119}$ ($^{113}$PYFRSLR$^{119}$) formed the α-helix H2, from which several conserved hydrophobic side chains protruded and formed multiple van der Waals contacts with the dimer pseudo-β-barrel (Fig. 3b, c). Specifically, the side chains of

Mmi1$^{Pro113}$, Mmi1$^{Tyr114}$, and Mmi1$^{Phe115}$ interacted with the dimer interface formed by the side chains of Erh1$^{Glu7}$, Erh1$^{Tyr74}$, Erh1$^{Arg79}$, Erh1$^{Val70}$, and Erh1$^{Pro83}$ (Fig. 3c). It is noteworthy that the hydrophobic residues involved in the dimer formation also contributed to the recognition of Mmi1$^{Trp112}$ and the α-helix H2. The importance of the hydrophobic interactions between Mmi1 H2 and the Erh1 dimer was confirmed by mutational analysis. Mmi1$^{Y114A}$ and Mmi1$^{F115A}$ showed reduced affinities for Erh1 as revealed by ITC (Supplementary Table 1).

**Fig. 3** Molecular interface between Erh1 and Mmi1$^{95-122}$. **a** Sequence alignment of *S. pombe* Erh1 and the enhancer of rudimentary homolog (ERH) proteins from *H. sapiens* (Human), *C. elegans* (Worm), and *D. melanogaster* (Fly). The alignment was generated by ESPript3 with CLUSTALW. The secondary structures of Erh1, as determined by DSSP, are shown above the sequences. The conserved residues at the ERH dimer interface are emphasized by up-pointing triangles below the sequences. The key residues for Mmi1$^{Trp1112}$ interaction are emphasized by down-pointing triangles above the sequences. Aligned amino acids in red share similar biophysical properties, those in white with red highlight are perfectly conserved, and blue boxes denote conserved clusters of amino acids. Black arrows denote hydrophobic residues, blue arrows denote residues involved in hydrogen bond interaction. **b** Sequence alignment of Mmi1$^{95-122}$ of *S. pombe* and the corresponding regions from *S. cryophilus*, *S. octosporus*, and *S. japonicus*. The secondary structures are also shown above the sequences. **c** Interactions of Mmi1$^{Trp112}$, α-helix H2, 3$_{10}$ helix H1, and N-terminal loop with Erh1 dimer. Erh1 (aquamarine and wheat) and Mmi1$^{95-122}$ (magenta) are shown as ribbons with selected side-chain and main-chain atoms as sticks. Hydrogen bonds are shown as black dashed lines. The Y97 and F99 subpanel shows the binding pocket for Mmi1$^{Tyr97}$ and Mmi1$^{Phe99}$; the Hydrogen bonding panel shows the hydrogen bonding network between Mmi1 and the Erh1 dimer; the H1 subpanel shows the interactions between the 3$_{10}$ helix H1 of Mmi1 EIM peptide and Erh1 dimer; the W112 subpanel shows the hydrophobic pocket of the Erh1 dimer for Mmi1$^{Trp112}$ recognition; the H2 subpanel shows the interactions between the α-helix H2 Mmi1 of the EIM peptide and the Erh1 dimer. **d** GST pull-down assay showing the diminished interaction between the Mmi1$^{W112A}$ mutant and Erh1. Source data are provided as a Source Data file. **e** ITC fitting curves of Erh1 using SUMO tagged Mmi1$^{95-122}$ (magenta) and Mmi1$^{W112A}$ (blue) are shown

Additional structures that contribute to EMC stability include the polypeptide chain $^{109}$SYE$^{111}$ that formed a short 3$_{10}$ helix H1 (Fig. 3c). The amide nitrogen and side-chain hydroxyl oxygen atoms of Mmi1$^{Ser109}$ formed bifurcated hydrogen bonds with the side chain of Erh1$^{Asp68}$. Consequently, the side chains of Mmi1$^{Tyr110}$ and Mmi1$^{Asp111}$ bulged out with no direct contact with Erh1. The conformation may allow Mmi1$^{Trp112}$ to point towards Erh1. Interestingly, the side chain of Mmi1$^{Tyr110}$ interacted with Mmi1$^{Arg119}$ via cation-π interactions. Furthermore, a hydrogen bond linked the carbonyl oxygen of Mmi1$^{Tyr110}$ and the side chain of Mmi1$^{Arg119}$. Confirming the importance of these interactions, the affinities of the Erh1$^{D68A}$ mutant for the Mmi1$^{95-122}$ peptide, and the Mmi1$^{R119A}$ mutant for Erh1, were reduced (Supplementary Table 1). We also observed that the H1 helix toward the amino-terminus, Mmi1 ($^{96}$KYDFSRHCTDYGH$^{108}$), adopted a long structural loop conformation featuring several turns, which was constrained by an intertwined network of both intra- and inter-molecular interactions. The inter-molecular interactions between Mmi1 and Erh1 contained both hydrophobic contacts and numerous direct hydrogen bonds. Most of the hydrophobic contacts involved Mmi1$^{Tyr97}$ and Mmi1$^{Phe99}$. The side chain of Mmi1$^{Tyr97}$ was surrounded by a shallow channel flanked by Erh1$^{Tyr42}$ and Erh1$^{Phe62}$. In the neighboring channel, the side chain of Mmi1$^{Phe99}$ was accommodated by Erh1$^{Ile14}$, Erh1$^{Tyr37}$, Erh1$^{Val41}$, Erh1$^{Phe62}$, and Erh1$^{Ile66}$ (Fig. 3c). Consistent with these observations, Mmi1$^{Y97A}$ and Mmi1$^{F99A}$ mutants showed reduced affinities for Erh1 (Supplementary Table 1).

Among the specific hydrogen bonds, most contacts involved Erh1$^{Asp27}$ and Erh1$^{Arg23}$ that also played a role in Erh1 dimer interactions. Across the β2 edge of the Erh1 monomer β-sheet, the carbonyl oxygen of Mmi1$^{Cys103}$ and the amide nitrogen of Mmi1$^{Asp105}$ formed hydrogen bonds with the main chain of Erh1$^{Asp27}$. Additionally, the side chains of Erh1$^{Asp27}$ and Erh1$^{Arg23}$ also formed numerous hydrogen bonds with Mmi1, including with the side chains of Mmi1$^{Trp112}$ and Mmi1$^{Thr104}$, as well as the carbonyl oxygen of Mmi1$^{Cys103}$ (Fig. 3c). Underscoring the importance of these Erh1 residues, substitution of Erh1$^{Asp27}$ or Erh1$^{Arg23}$ with alanine significantly reduced the binding affinity of Erh1 for Mmi1$^{95-122}$ (Supplementary Table 1). Furthermore, the side chain and the carbonyl oxygen of Erh1$^{Tyr67}$ formed hydrogen bonds with the carbonyl oxygen of Mmi1$^{Gly107}$ and the Nε2 group of Mmi1$^{H108}$, respectively. Hydrogen bonds also formed between Mmi1$^{Glu98}$ and Mmi1$^{Asp105}$ and the side chains of Erh1$^{Thr44}$ and Erh1$^{Ser33}$, respectively (Fig. 3c). In addition to these inter-molecular interactions, there are several intra-molecular interactions that helped constrain the structural loop conformation of the Mmi1 peptide: two hydrogen bonds between the side chain of Mmi1$^{Asp98}$ and the side chain of Mmi1$^{Ser100}$; one hydrogen bond between the side chain of

Mmi1$^{Thr104}$ and the nitrogen atom of Mmi1$^{Tyr106}$ (Fig. 3c). Overall, this complex network of interactions serves to stabilize the structure of Mmi1 and contributes to EMC formation.

**The Erh1-Mmi1 binding interface is conserved in human ERH.** We next investigated whether the Erh1 dimer interface that interacts with Mmi1 shares features with human ERH. In addition to conservation of residues at the dimer interface (Fig. 3a), structural comparison of the Mmi1-bound Erh1 homodimer with the previously characterized human ERH homodimer (PDB ID:1WZ7) revealed significant similarities (with a r.m.s deviation of 0.98 Å over 135 Cα atoms) (Supplementary Fig. 3a). Moreover, the conformations of the residues at the dimer interface were perfectly conserved. The Erh1 binding pocket that mediates binding to Mmi1$^{Trp112}$ was also nearly identical between *S. pombe* Erh1 and human ERH (Supplementary Fig. 3b, c). This high degree of structural conservation between evolutionarily distant ERH family proteins suggests that the dimer interface might be a conserved protein–protein interaction platform that facilitates binding of ERH family proteins to species-specific factors.

**The Erh1-Mmi1 interface is required for EMC function in vivo.** Guided by the results from our structural and biochemical analyses above, we created a mutant *mmi1* allele to investigate the biological significance of EMC assembly (Supplementary Fig. 4a). We reasoned that an *S. pombe* strain expressing an *mmi1$^{W112A}$* mutant allele, which should disrupt the Erh1-Mmi1 interface and prevent EMC assembly, would allow us to uncouple EMC functions from other roles specific to either Mmi1 or Erh1. The mutant Mmi1$^{W112A}$ protein was expressed at levels comparable to that of the WT Mmi1 (Fig. 4a and Source Data). However, replacing WT Mmi1 with Mmi1$^{W112A}$ drastically diminished the Erh1-Mmi1 interaction as determined by co-immunoprecipitation (Fig. 4b). This finding is consistent with the conclusion that Mmi1$^{Trp112}$ is critical for EMC assembly in vivo.

We next compared the phenotypes of *mmi1$^{W112A}$* and *erh1Δ*. Cells expressing the mutant protein showed no obvious growth defects at 32 °C and 37 °C (Fig. 4c). However, *mmi1$^{W112A}$* cells displayed cold-sensitivity at 18 °C, similar to *erh1Δ* (Fig. 4c). We also tested whether mating and meiotic progression were affected by exposing WT and mutant colonies to iodine vapor[38]. When meiosis is induced, homothallic WT cells sporulate and form asci. The starch-like compound in the spore wall is stained with dark color when exposed to iodine vapor. Indeed, *mmi1$^{W112A}$* cells showed decreased intensity of iodine staining and a reduced mating efficiency, in a manner similar to *erh1Δ* (Supplementary Fig. 4b). Thus, *mmi1$^{W112A}$* mimics the defective mating and meiotic progression phenotypes displayed by cells lacking Erh1.

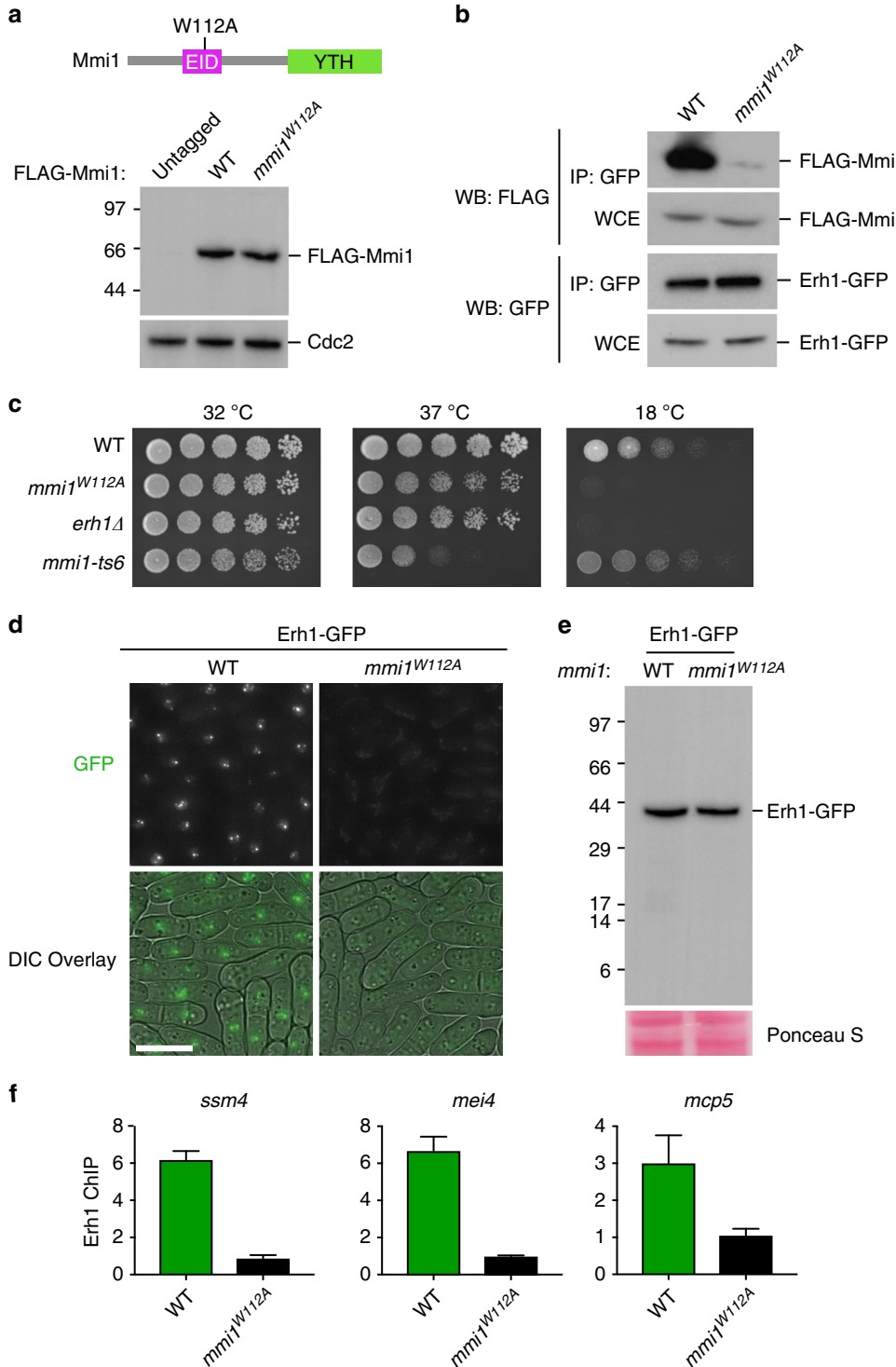

**Fig. 4** Mmi1[W112A] perturbs EMC functions in vivo. **a** Western blot analysis of FLAG-Mmi1 WT and FLAG-Mmi1[W112A] protein expression levels. Cdc2 serves as loading control. Also shown is a linear schematic of *mmi1[W112A]* mutant. Source data are provided as a Source Data file. **b** Coimmunoprecipitation of Erh1-GFP and FLAG-Mmi1 WT or FLAG-Mmi1[W112A]. Source data are provided as a Source Data file. **c** Spotting assays at 32, 37, or 18 °C on non-selective media. The *mmi1-ts6* strain was included as a control for 37 °C growth. **d** Erh1 localization in WT or *mmi1[W112A]* cells. WT and mutant cells expressing Erh1-GFP were imaged using a DeltaVision Elite fluorescence microscope (Applied Precision, GE Healthcare). Scale bar (white) represents 5 μm. **e** Western blot analysis depicting Erh1 protein levels in wild-type or *mmi1[W112A]* cells. Ponceau S stain serves as a loading control. Source data are provided as a Source Data file. **f** ChIP-qPCR analyses of Erh1-GFP enrichment in WT and *mmi1[W112A]* cells at indicated loci. Fold enrichment values plotted were calculated relative to the control *leu1* locus. Shown are mean±SD for two experiments

We further explored whether Mmi1$^{W112A}$ affects the localization of Erh1. Erh1 forms nuclear foci that co-localize with RNA elimination factors including Mmi1[14]. Indeed, Erh1 foci were not observed when GFP-tagged Erh1 (Erh1-GFP) was expressed in mmi1$^{W112A}$ mutant cells (Fig. 4d). This was not due to destabilization of Erh1, as comparable levels of Erh1-GFP were detected in WT and mmi1$^{W112A}$ (Fig. 4e). Furthermore, localization of Erh1 at EMC target loci was abolished in mmi1$^{W112A}$ (Fig. 4f). Taken together, these results suggest that EMC assembly is critical for Erh1 association with its target loci, and that the Mmi1$^{W112A}$ mutation specifically disrupts EMC assembly without affecting other essential function(s) of Mmi1.

**Erh1-Mmi1 interaction is required for H3K9me at islands.** RNA-based mechanisms target Clr4 H3K9 methyltransferase to promote heterochromatin assembly[39]. In addition to constitutive heterochromatin domains at centromeres, telomeres, and the mating-type locus, discrete blocks of facultative heterochromatin islands are detected across the S. pombe genome[40]. Among these, the assembly of heterochromatin islands at meiotic genes requires RNA elimination machinery[19,26–28]. Indeed, cells lacking Mmi1, which binds to DSR-containing meiotic mRNAs, or Erh1 are defective in the assembly of meiotic heterochromatin islands[14,19,26–28,34].

We utilized the mmi1$^{W112A}$ to explore whether the Mmi1-Erh1 interaction, rather than the individual functions of these factors, is critical for facultative heterochromatin assembly. A comparison of the genome-wide distribution of H3K9me in mmi1$^{W112A}$ and WT cells revealed a dramatic reduction in H3K9me at many facultative heterochromatin islands in the mutant (Fig. 5a, b and Supplementary Fig. 5a, b). Defects in H3K9me in mmi1$^{W112A}$ occurred specifically at meiotic heterochromatin islands that also require Erh1 for heterochromatin assembly (Supplementary Table 3)[14]. H3K9me at Erh1-independent heterochromatin islands was not affected in mmi1$^{W112A}$ (Fig. 5c and Supplementary Fig. 5a, b). Moreover, H3K9me at constitutive heterochromatin domains, such as at pericentromeric regions and telomeres, was not affected in mmi1$^{W112A}$ or erh1Δ (Fig. 5a and Supplementary Fig. 5c). Conventional ChIP combined with real-time quantitative PCR (ChIP-qPCR) confirmed defects in H3K9me in mmi1$^{W112A}$ at Erh1-dependent islands associated with the meiotic genes ssm4 and mei4 (Supplementary Fig. 5d). Overall, these results suggest that the Erh1-Mmi1 interaction to form EMC is required for the assembly of facultative heterochromatin islands at meiotic genes.

**Erh1-Mmi1 interaction is required for gene silencing.** In addition to the assembly of heterochromatin islands, Mmi1 and Erh1 promote mRNA decay to prevent the untimely expression of gametogenic genes in vegetative cells[14]. Indeed, aberrant expression of meiotic genes is a major cause of chromosomal abnormalities associated with cancer and other diseases[41]. To investigate whether mmi1$^{W112A}$ affects any of the transcripts that are upregulated in mmi1Δ and/or erh1Δ cells, we compared the transcriptomes of WT and mutant cells using RNA-seq. The strains carried a non-functional truncated allele of mei4 that rescues lethality caused by the deletion of mmi1[18]. Interestingly, the expression profiles of mmi1$^{W112A}$ and erh1Δ were strikingly similar (Fig. 6a). The increase in transcripts observed in mmi1$^{W112A}$ was comparable to that observed in erh1Δ (Fig. 6b). Of the 138 transcripts upregulated in either mmi1$^{W112A}$ or erh1Δ (≥2-fold), a major class of transcripts was derived from loci that showed increased expression during nitrogen starvation and meiotic cell cycle progression. These included DSR-containing meiotic mRNAs and various lncRNAs that are targeted for degradation by RNA elimination factors[19,35,36]. The observed upregulation of DSR-containing mRNAs derived from ssm4, rec8, spo5, and mcp5 in mmi1$^{W112A}$ was confirmed using real-time quantitative PCR (RT-qPCR) (Supplementary Fig. 6a). These results show that mmi1$^{W112A}$ phenocopies erh1Δ and support the idea that the binding of Erh1 to Mmi1 is a prerequisite for EMC-dependent mRNA decay and gene silencing.

The majority of transcripts that showed elevated levels in mmi1$^{W112A}$ and erh1Δ were also upregulated in mmi1Δ (Fig. 6a). However, the levels of upregulated transcripts were generally higher in mmi1Δ cells as compared to mmi1$^{W112A}$ and erh1Δ cells (Fig. 6a and Supplementary Fig. 6b). More importantly, we found that the loss Mmi1 caused upregulation of a large number of coding and non-coding RNAs that were not affected by mmi1$^{W112A}$ or erh1Δ (Fig. 6a, c). We conclude from these analyses that the association between Mmi1 and Erh1 is critical for controlling a specific subset of the Mmi1 regulon, and that Mmi1 regulates gene expression in both EMC-dependent and -independent manners.

**EMC prevents nuclear export of gametogenic gene transcripts.** Mmi1 tethers gametogenic gene transcripts to nuclear foci to prevent their translation and expression in mitotically dividing cells[42]. We wondered whether EMC assembly rather than the individual Mmi1 or Erh1 protein is required for this nuclear sequestration activity. To test this, we assayed the localization of a DSR-containing ssm4 mRNA in mmi1$^{W112A}$ mutant cells that express Mmi1 and Erh1 at levels comparable to wild-type cells but are defective in EMC assembly. Single molecule RNA fluorescence in situ hybridization (smFISH) revealed ssm4 mRNA predominantly localizing to specific nuclear foci in WT, whereas cytoplasmic accumulation of ssm4 mRNAs was observed in mmi1Δ mutant cells (Fig. 7a, b) as observed previously[42]. Interestingly, ssm4 mRNAs were exported into the cytoplasm of mmi1$^{W112A}$ mutant cells (Fig. 7a, b), suggesting that EMC formation is critical for nuclear retention of DSR-containing transcripts to prevent their untimely expression.

**Mmi1$^{W112A}$ uncouples EMC-dependent and -independent functions.** Mmi1 also promotes alternative polyadenylation and transcription termination of meiotic mRNAs and regulatory lncRNAs[34–36]. For example, binding of Mmi1 to the ssm4 transcript triggers pre-mature pre-mRNA 3′-end formation near regions containing DSR elements[34]. This process, which involves the RNA polymerase II termination factor Dhp1/Xrn2 and the nuclear exosome Rrp6, precludes utilization of the canonical polyadenylation (polyA) signal further downstream[34]. Similarly, elegant studies have shown that Mmi1 selectively promotes termination of lncRNAs involved in developmental and environmental control of gene expression. Specifically, Mmi1 mediates transcription termination to prevent lncRNAs from invading and repressing downstream genes[35,36].

In light of these previous studies, we wondered whether Mmi1 controls polyA site selection as part of EMC. We utilized the mmi1$^{W112A}$ mutant, in which the Mmi1 mutant protein and Erh1 are expressed at levels comparable to those in WT but fail to form the EMC. 3′-RACE at the ssm4 locus showed that pre-mature 3′-end processing was abolished in mmi1Δ, resulting in full-length mRNA terminating at the canonical polyA site (Fig. 8a and Source Data). In contrast, utilization of the canonical polyA signal was not observed in mmi1$^{W112A}$, which instead displayed usage of cryptic sites in regions containing multiple hexameric DSR elements (Fig. 8a). An identical pattern was observed in erh1Δ (Fig. 8a). These results suggest that EMC assembly is not essential for preventing the use of the canonical polyA signal.

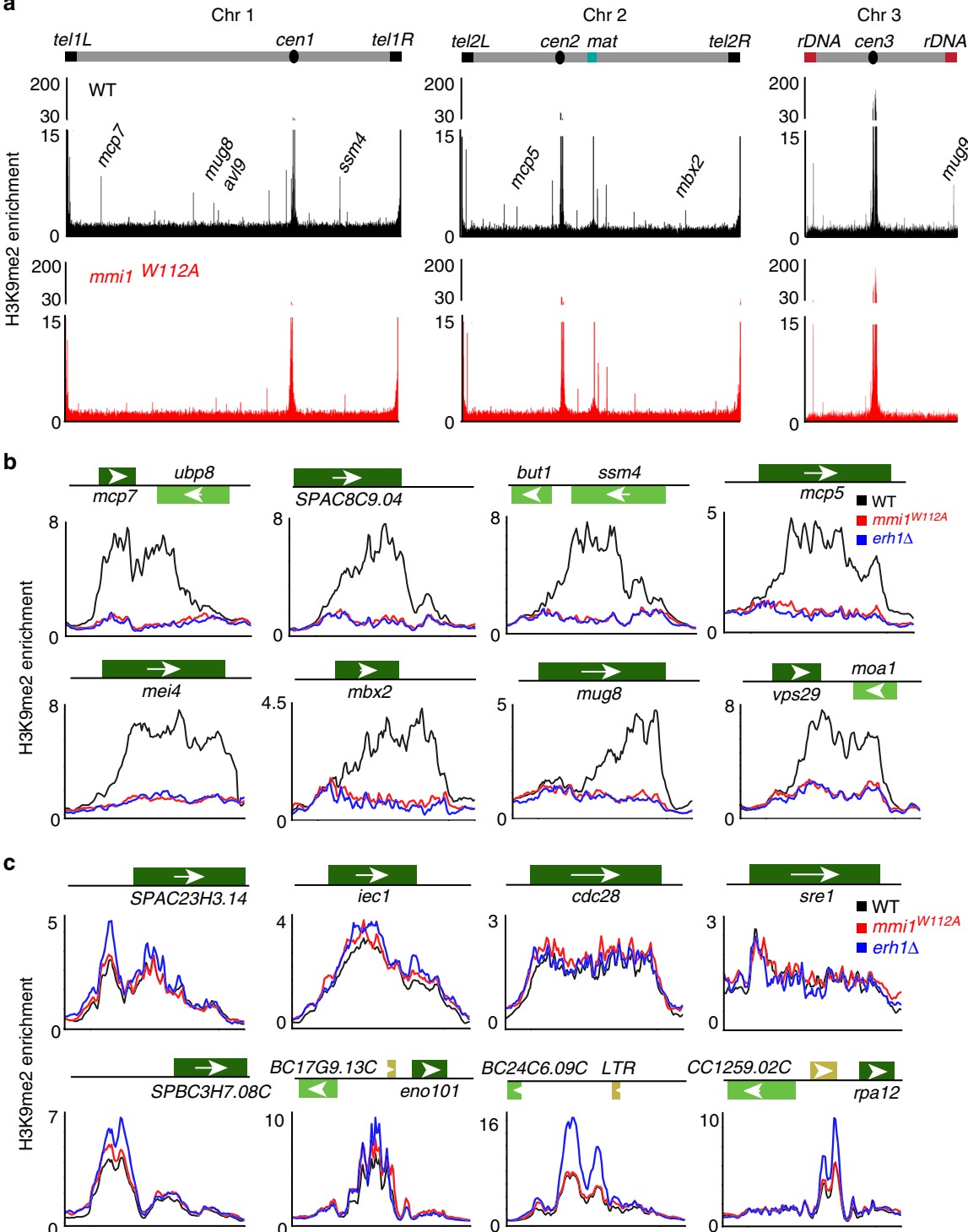

**Fig. 5** Mmi1[W112A] affects EMC-dependent facultative heterochromatin islands. **a** Genome-wide ChIP-seq profiles showing enrichment of histone H3 dimethylation (H3K9me2) in WT and *mmi1[W112A]* cells. Examples of several individual heterochromatin islands are indicated. **b** H3K9me2 enrichment ChIP-seq profiles of EMC-dependent islands. **c** H3K9me2 enrichment ChIP-seq profiles of EMC-independent islands. The color scheme used in all plots shown is WT (black), *mmi1[W112A]* (red), and *erh1Δ* (blue)

We next wondered whether assembly of EMC is also dispensable for the role of Mmi1 in selective termination of lncRNAs. Notably, RNA-seq analyses revealed that lncRNAs targeted by Mmi1 extended into downstream genes in *mmi1Δ*, but not in *mmi1[W112A]* or *erh1Δ* (Fig. 8b). Northern analyses of *prt* and *nam1* lncRNAs, which regulate expression of downstream *pho1* and *byr2* genes, respectively[19,35,36,43], revealed defective termination of lncRNAs in *mmi1Δ*, but not *mmi1[W112A]* or *erh1Δ* (Fig. 8c, d). Read-through transcripts resulting from defects in termination of *prt* and *nam1* were specifically detected in the *mmi1Δ* mutant (Fig. 8c, d). These longer transcripts (named *prt-L* and *nam1-L*) were not detected in *mmi1[W112A]* or *erh1Δ* (Fig. 8c, d). Defective termination in *mmi1Δ* but not *mmi1[W112A]* or *erh1Δ* was also observed at other loci (Supplementary Fig. 7). These

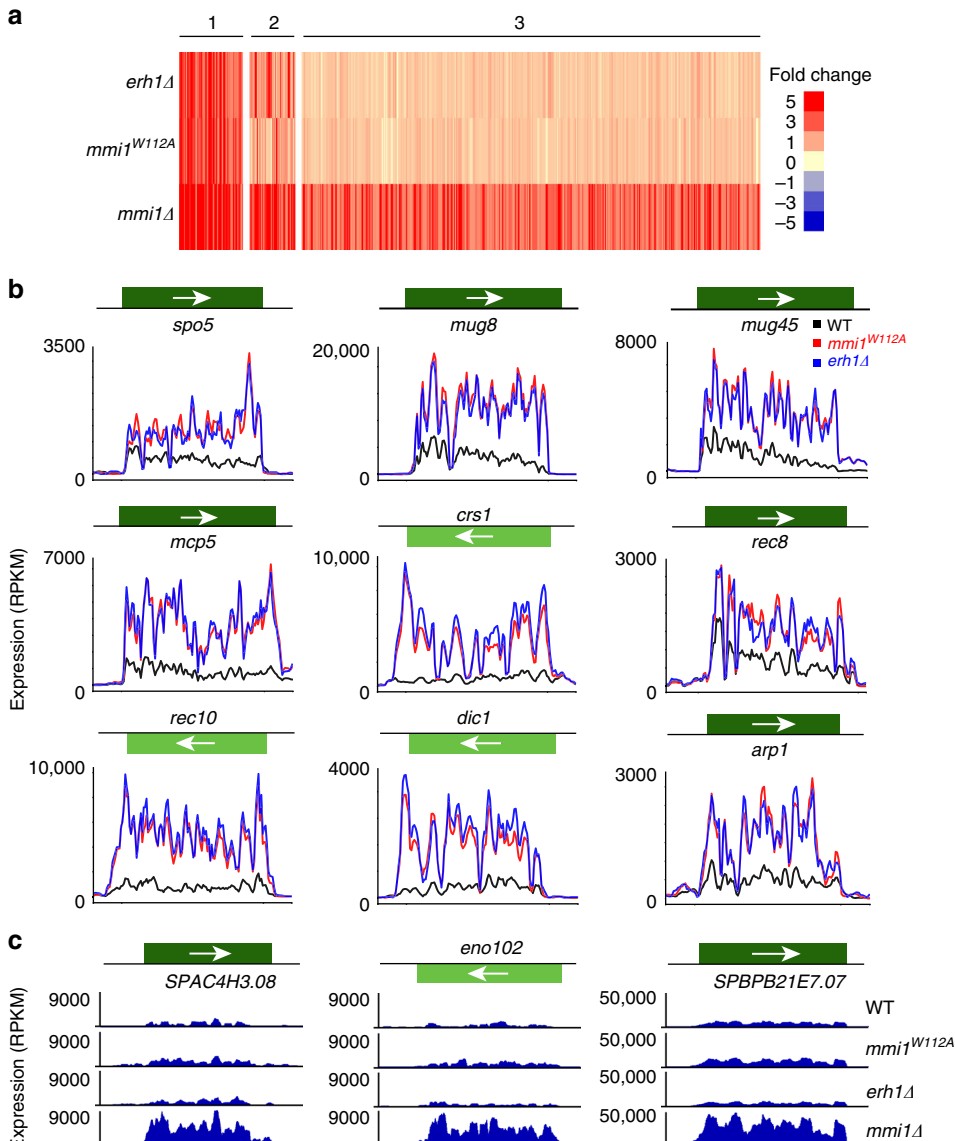

**Fig. 6** Mmi1<sup>W112A</sup> affects a subset of the Mmi1 regulon silenced by EMC. **a** Heatmap of genes upregulated by ≥2-fold in at least one mutant as compared to WT. Cluster 1 contains genes that are derepressed in *mmi1Δ*, *mmi1*<sup>W112A</sup>, and *erh1Δ*. Cluster 2 contains genes that are derepressed in *mmi1Δ* and *mmi1*<sup>W112A</sup> or *erh1Δ*. Cluster 3 contains genes that are derepressed in *mmi1Δ* only. In total, there are 531 genes considered here as upregulated out of a possible 7019. Source data are provided as a Source Data file. **b** RNA-seq profiles of 9 representative genes whose repression requires Erh1-Mmi1 interaction. The color scheme used is WT (black), *mmi1*<sup>W112A</sup> (red), and *erh1Δ* (blue). Shown is normalized RPKM expression. **c** RNA-seq profiles of 3 representative loci showing derepression in *mmi1Δ* but not in *mmi1*<sup>W112A</sup> and *erh1Δ* cells

results suggest that Mmi1 promotes termination of regulatory lncRNAs via a mechanism independent of its interaction with Erh1 or EMC formation.

Our analyses using *mmi1*<sup>W112A</sup> also uncovered an EMC-dependent function of Mmi1 in the repression of *pho1* mRNA, a function obscured in *mmi1Δ* cells, which are defective in both EMC assembly and in the termination of *prt* lncRNA involved in *pho1* repression. Unlike *mmi1Δ* cells that show only a small change in *pho1* gene expression, *mmi1*<sup>W112A</sup> cells showed a marked increase in the level of *pho1* mRNA (Fig. 8b, c). A similar increase was also observed in *erh1Δ* cells (Fig. 8b, c). These results suggest that Mmi1 represses *pho1* expression as part of EMC. However, this function of Mmi1 is obscured in *mmi1Δ* cells (Fig. 8b). This is due to defective termination of *prt* lncRNA that permits the long *prt* transcript to invade and repress *pho1* (Fig. 8c top panel). Indeed, deletion of *prt* alleviated the repression of

*pho1* observed in *mmi1Δ* cells (Fig. 8c bottom panel). Similar changes are detected at other loci such as the *SPCC11E10.01* gene, which is also repressed by read-through lncRNA in *mmi1Δ* cells (Supplementary Fig. 7). Thus, the Mmi1<sup>W112A</sup> mutant that affects Mmi1-Erh1 interaction without disrupting the termination function of Mmi1 provides a unique tool to uncouple EMC-dependent and -independent functions of Mmi1.

## Discussion

This study describes the structural and functional analysis of the nuclear RNA processing complex EMC. EMC contains the evolutionarily conserved ERH family protein Erh1 and the YTH-domain RNA-binding protein Mmi1, which prevent the deleterious effects of untimely gametogenic gene expression in mitotic cells[9,14,18,41] (this study). Erh1 interacts with Mmi1 through an interface that is conserved in the human ERH protein. The

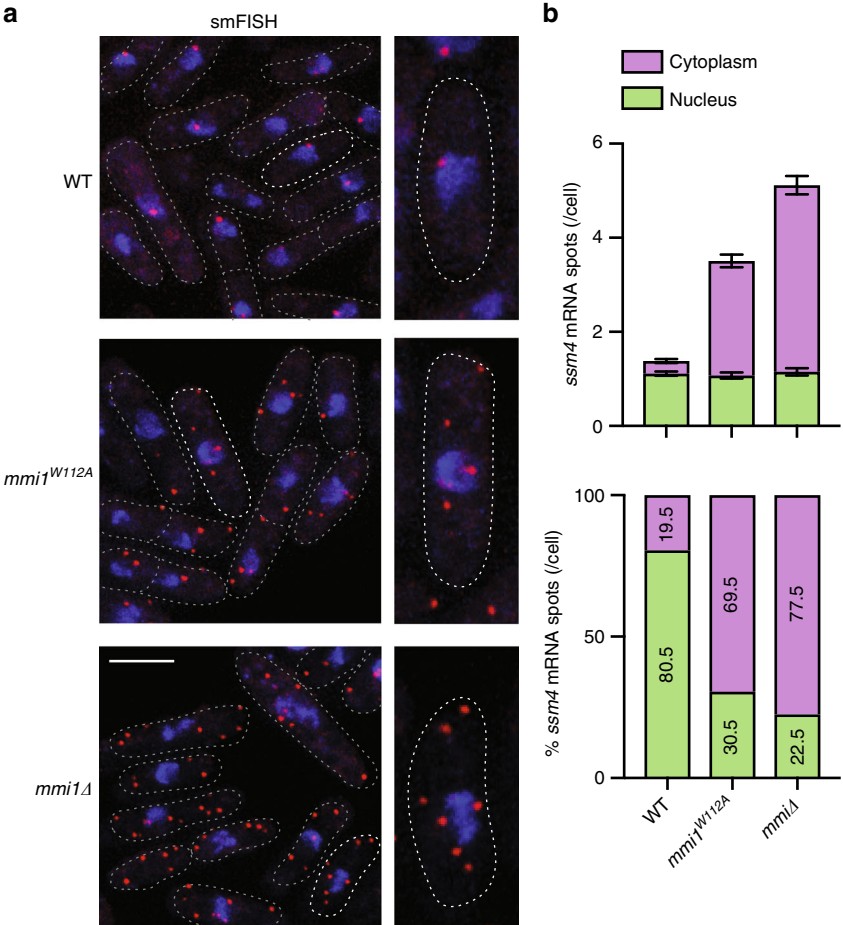

**Fig. 7** EMC assembly is critical for nuclear retention of gametogenic gene transcripts in mitotic cells. **a** Representative images of the EMC target *ssm4* mRNA (red) detected by Single molecule RNA Fluorescence In-Situ Hybridization (smFISH). DNA was stained with DAPI (blue). Images are shown as the maximum-intensity projections of Z-stacks. Dotted lines indicate the outline of cells. Scale bar (white) represents 5 μm. **b** Quantification of nuclear and cytoplasmic localization of *ssm4* mRNA. The upper panel shows the mean ± SEM from more than 140 cells and the lower panel indicates the distribution of the percentages of *ssm4* mRNA spots/cell

structure-based design of a mutant allele that disrupts Mmi1 binding to the Erh1 interface enabled us to determine the biological significance of EMC assembly. Ultimately, we uncovered a specific requirement for EMC among the various functions attributed to Mmi1 including RNA-mediated heterochromatin assembly, nuclear retention of transcripts, and gametogenic gene silencing.

Erh1 associates with well-known RNA processing activities such as CCR4-NOT and MTREC that are linked to mRNA decay and assembly of facultative heterochromatin domains at specific genomic loci[14,19,22,25,27,44]. The targeting of Erh1 and its associated factors requires Mmi1[14], which contains two notable domains, including a carboxy-terminal YTH domain implicated in binding to DSR-containing RNAs[18,29–33] and a conserved amino-terminal domain that is largely unstructured[30](this study). Our analyses show that the region containing the YTH domain of Mmi1 is dispensable for its interaction with Erh1. Instead, it is the amino-terminal domain of Mmi1 (residues 95–122) that interacts with Erh1 to form EMC. These observations suggest that Mmi1-mediated degradation of RNA and heterochromatin formation require cooperation between its carboxy- and amino-terminal domains. Whereas Mmi1 binds target RNAs via its YTH domain, it engages Erh1 and RNA processing complexes via its amino-terminal domain (Supplementary Fig. 8).

Our biochemical and co-crystal analyses show that EMC is a heterotetrameric complex, wherein Mmi1 is bound to the surface of the Erh1 dimer interface with 2:2 stoichiometry. In particular, we found that Mmi1 adopts a mixed loop-helix conformation when bound to Erh1. Intra and inter-molecular interactions between Mmi1 and Erh1 are driven by hydrophobic contacts and are supported by hydrogen bonds, with residue Mmi1$^{Trp112}$ playing a crucial role in the formation or stabilization of EMC. Indeed, our in vitro and in vivo analyses showed that mutation of the Mmi1$^{Trp112}$ affected its interaction with Erh1, hence compromising the assembly of EMC (Figs. 3d, 4b, and Supplementary Table 1). In light of the structural features of EMC, it is possible that incorporation of Mmi1 into EMC facilitates dimerization, a feature critical for nuclear sequestration of meiotic mRNAs[42]. Consistent with this possibility, we note that a domain (residues 61–180) reported to be important for Mmi1 self-association[42] overlaps with the Mmi1 amino-terminal domain (residues 95–122) and our analyses show that EMC formation is critical for nuclear retention of meiotic mRNAs.

Another major finding from our analyses is the highly conserved molecular interface on the Erh1 dimer to which Mmi1 binds. Erh1 homodimers bound to Mmi1 show strong similarities with human ERH homodimers (this study)[37,45]. The high conservation of the binding pocket and dimerization interface is surprising because, at the primary sequence level, the evolutionarily distant Erh1 and ERH share <30% amino acid identity. This structural conservation suggests that ERH in other species

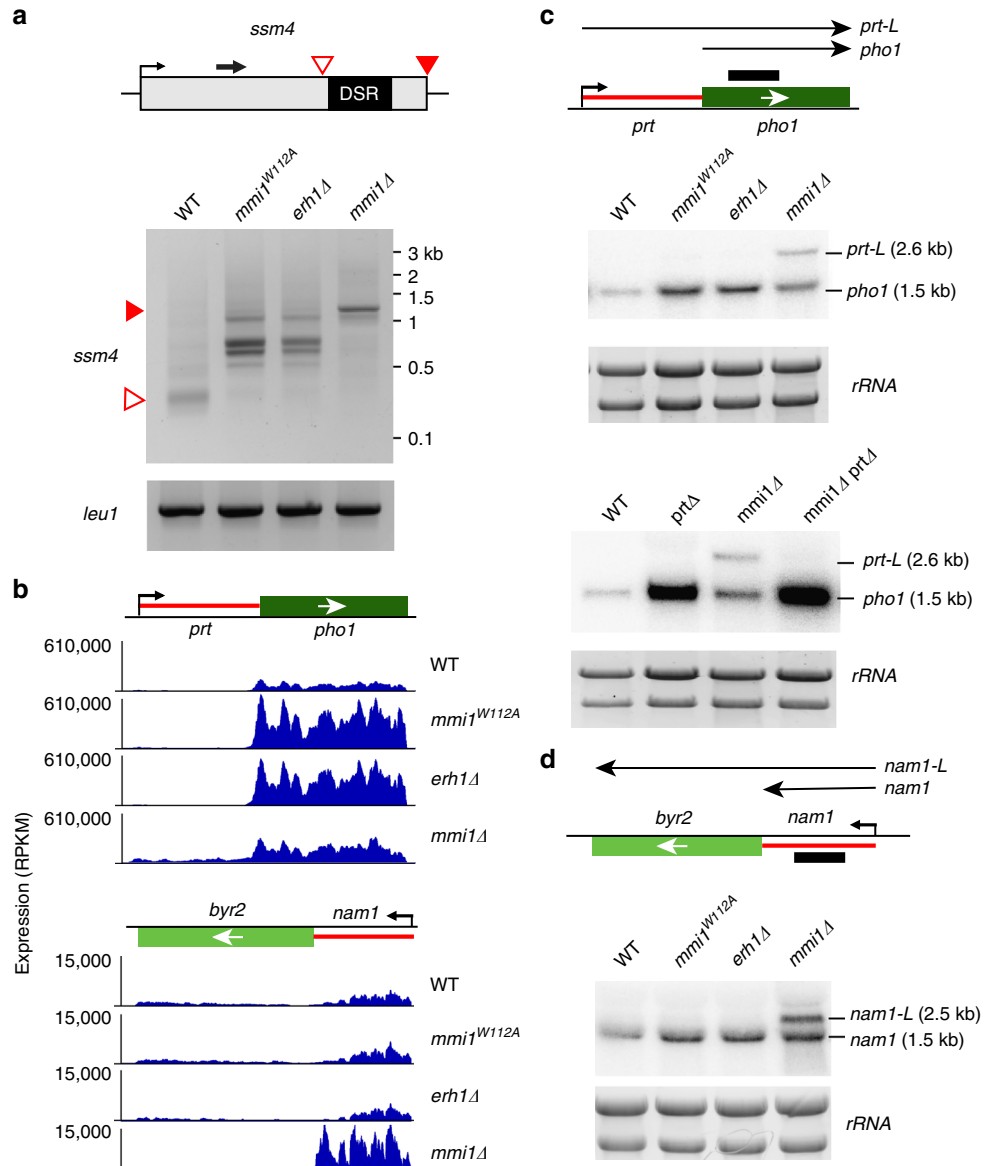

**Fig. 8** Mmi1^W112A uncouples EMC-dependent and -independent Mmi1 functions. **a** 3′ RACE of *ssm4* and *leu1* (control) loci. Red triangles indicate termination sites as determined by sequencing of 3′ RACE products. The black arrow indicates the gene-specific forward primer used, which can be found in Supplementary Table 4. **b** RNA-seq expression profiles of *prt-pho1* loci and *nam1-byr2* in WT, *mmi1^W112A*, *erh1Δ*, and *mmi1Δ*. Shown is normalized RPKM expression. **c** Northern blot analysis at the *prt-pho1* locus in indicated strains using a probe targeting *pho1* mRNA (top and bottom panels). **d** Northern blot analysis at the *nam1-byr2* locus in indicated strains using a probe targeting *nam1* ncRNA. Source data are provided as a Source Data file

may use the same interface to interact with its binding partners to regulate expression of specific target genes.

Importantly, structure-guided design of a mutant allele enabled us to dissect the role of EMC in various functions ascribed to Mmi1. Although several activities have been reported for Mmi1[18,23,26–28,34–36,42], which of them require association of Mmi1 with Erh1 to form EMC, and which require only the individual protein, remained to be elucidated. Because the *mmi1^W112A* mutant disrupted Mmi1-Erh1 without affecting protein levels, we were able to discover that the Mmi1-Erh1 interaction is indeed required for RNA-mediated assembly of heterochromatin islands targeting meiotic genes. Moreover, EMC assembly is required for silencing a specific subset of the entire Mmi1 regulon. Surprisingly, EMC is dispensable for Mmi1-dependent premature pre-mRNA 3′-end formation of meiotic transcripts and termination of regulatory lncRNAs. Indeed, only

the strain carrying a deletion of *mmi1*, and not the mutant defective in EMC assembly (*mmi1^W112A*), showed defects in termination of target transcripts. This finding suggests that Mmi1 also acts independently to engage additional factors involved in termination and gene regulation.

The ability to specifically disrupt EMC in the *mmi1^W112A* mutant, without interfering with the termination function of Mmi1, provided insights into gene regulation that could not be revealed using the *mmi1Δ* null allele. For example, Mmi1 participates in overlapping mechanisms to control expression of genes located adjacent to cis-acting regulatory lncRNA (such as *pho1*). As part of EMC, which is associated with RNA processing activities, Mmi1 represses *pho1* mRNA. Mmi1 also acts independently of EMC to modulate *pho1* expression by mediating transcription termination of the upstream lncRNA, thus preventing it from invading and repressing *pho1*. Indeed, the strong

upregulation of *pho1* observed in *mmi1*[W112A] cells, which are defective only in EMC assembly, is obscured in *mmi1Δ* cells by the invasion of lncRNA defective in Mmi1-mediated termination (Fig. 8b, c). The interplay between overlapping Mmi1-dependent mechanisms at environmentally or developmentally regulated loci remains to be determined, but it is plausible that such a system facilitates rapid fine-tuning of gene expression during abrupt changes. Whether Mmi1 affects the loading of gene silencing factors such as HDACs implicated in lncRNA-mediated silencing of genes is also unknown[46,47].

Collectively, we provide important insight into EMC and open up other avenues for investigating conserved ERH and YTH family proteins[16,48,49]. Structural similarities and the highly conserved Erh1 interaction interface from *S. pombe* to human suggest that ERH family members may all contain a structurally conserved scaffold for connecting with species-specific proteins to mediate diverse cellular functions. Since protein–protein interactions are primarily driven by structural attributes, the ERH interface may be evolutionarily conserved due to its ability to recognize intrinsically disordered regions (IDRs) crucial for the functions of various RNA-binding proteins[50]. As RNA-binding proteins have evolved to recognize species-specific RNA substrates, ERH family members might provide a conserved link between these RNA-binding proteins and RNA-processing effectors. Further probing of the molecular architecture and functions of the ERH and YTH proteins that control several different aspects of RNA metabolism is crucial for understanding how these proteins prevent aberrant gene expression and chromosomal abnormalities[11,41], and may reveal therapeutic targets for treatment of human diseases.

## Methods

**Protein expression and purification**. A region of Erh1 (residues 1–104) was amplified by PCR from the *S. pombe* genome and cloned into a modified pET28a (Novagen) vector without a thrombin protease cleavage site (p28a). GST-Mmi1[YTH] was sub-cloned from a Mmi1[YTH]-p28a construct to a modified PGex-4T1 vector with a TEV protease cleavage site (Tev4T1). A region of Mmi1 (residues 1-122) was amplified by PCR from the *S. pombe* genome and cloned into the Tev4T1 vector. GST-tagged Mmi1[65–122], Mmi1[95–122], Mmi11[95–111], and Mmi1[102–111] were generated from the Mmi1[1–122]-Tev4T1 construct by a MutanBEST kit (Takara). SUMO tagged Mmi1[95–122] was sub-cloned from the Mmi[95–122]-Tev4T1 construct to a modified pET28a vector with a SUMO protein fused at the N-terminus following the His$_6$ tag. Mutants were generated using the Takara MutanBEST kit. The Erh1-(Gly-Ser-Ser)$_5$-Mmi1[95–122] fusion clone was constructed by overlap extension PCR and was also cloned to the p28a vector. All proteins were expressed in *Escherichia coli* BL21 (Gold) cells. Cells were grown in LB medium at 37 °C until the OD$_{600}$ reached about 0.8. Protein expression was induced with 0.1 mM β-D-1-thiogalactopyranoside (IPTG) for 24 h at 16 °C. The His$_6$-tagged proteins, as well as SUMO tagged Mmi1 peptides, were purified by Ni-chelating resin (Qiagen) in a buffer containing 20 mM Tris-HCl (pH 8.0), 500 mM NaCl. The GST tagged proteins were purified by glutathione sepharose (GE healthcare) in a buffer containing 20 mM Tris-HCl (pH 7.5), 500 mM NaCl. Proteins were further purified by size-exclusion chromatography on a Hiload 16/60 Superdex 75 column (GE healthcare) in a buffer containing 20 mM Tris-HCl (pH 7.5), 150 mM NaCl.

**Crystallography**. The crystals were grown using the hanging drop vapor diffusion method at 20 °C by mixing 1 μl Erh1-(Gly-Ser-Ser)$_5$-Mmi1[95–122] fusion protein (10 mg per mL) with 1 μl reservoir buffer (1.6 M (NH$_4$)$_2$SO$_4$, 0.1 M MES pH 6.5). The reservoir solution supplemented with 25% glycerol was used as a cryoprotectant. The X-ray diffraction data set was collected on beamline 18U1 at Shanghai Synchrotron Radiation Facility (SSRF BL18U1). The data set was indexed and integrated by iMosflm and scaled by SCALA in CCP4i suite[51,52]. The initial crystallographic phases were calculated using molecular replacement that was carried out by Phaser employing the structure of human ERH protein (PDB ID: 1WZ7) as the search model[53]. An initial model was automatically built by Buccaneer[54]. The model was further built and refined using Coot and Phenix, respectively[55,56].

**Isothermal titration calorimetry**. ITC assays were carried out on a MicroCal iTC200 calorimeter (GE Healthcare) at 293 K. Because a synthesized Mmi1[95–122] peptide could not be dissolved in the interaction buffer (20 mM Tris-HCl pH 7.5, 150 mM NaCl), and the protein construct corresponding to Mmi1[95–122] precipitated upon removal of the solubility tag, likely due to a high proportion of hydrophobic residues in the sequence, we conducted the ITC experiments using SUMO-tagged Mmi1 peptides for titration into Erh1. The titration protocol

consisted of a single initial injection of 1 μl, followed by 19 injections of 2 μl SUMO-tagged Mmi1 peptides into the sample cell containing Erh1 protein. Thermodynamic data were analyzed with a single-site binding model using MicroCal PEAQ-ITC Analysis Software provided by the manufacturer.

**Strains and media**. Standard yeast culturing and genetic manipulation methods were used. *S. pombe* strains used in this study are listed in Supplementary Table 4. Strains carrying the *prtΔ* allele contain a deletion of the region −400 to −1200 bp upstream of *pho1*[19]. All in vivo experiments were performed in yeast extract rich medium supplemented with adenine (YEA) at 18 °C, 32 °C, or 37 °C, as indicated.

**Construction of Mmi1[W112A] *S. pombe* strains**. The untagged *mmi1*[W112A] strain was constructed using the approach outlined in Supplementary Fig. 4a. The 5′ region of *mmi1* and an upstream homology region were amplified so that the W112A mutation was designed as part of the reverse primer with 80 bp downstream homology region. The resultant amplicon was transformed into a strain containing *ura4+* inserted adjacent to *mmi1*. Transformants in which the *ura4+* was popped-out by the PCR amplicon were identified using counterselective FOA medium. Sanger sequencing was used to verify the presence of the W112A mutation.

The *flag-mmi1*[W112A] strain was constructed by first amplifying the *mmi1*[W112A] coding region from the untagged strain described above. The resultant amplicon was co-transformed with pREP3X plasmid, carrying a *LEU2* selection marker, into a *flag-mmi1* (wild-type *mmi1*) strain. Transformants were selected on medium lacking leucine at 32 °C to obtain single colonies. Single colonies were replica plated onto rich YEA medium plates and further incubated at 18 °C or 32 °C. Transformants carrying *flag-mmi1*[W112A] were identified based on their poor growth at 18 °C. Sanger sequencing of the *mmi1* gene amplified from the mutant strain was used to verify the presence of the mutant allele. Western blotting was performed to confirm protein expression levels, as shown in Fig. 4a.

**Immunoprecipitation and Western blotting**. Cells were grown to mid-log phase in rich YEA medium at 32 °C, harvested, and flash-frozen in liquid nitrogen prior to extract preparations. Extracts prepared from yeast cells expressing epitope tagged proteins under the control of native gene promoters were used for immunoprecipitations and Western blot analyses as described[28]. Total protein extracts for western blotting analyses were prepared by trichloroacetic acid (TCA) precipitation. Briefly, cells were lysed using glass beads in 20% TCA buffer. Next, lysates were diluted using 5% TCA buffer and total protein pellets were concentrated by centrifugation at 21,000×g for 5 min. Precipitated protein was dissolved in sodium dodecyl sulfate (SDS) sample buffer prior to resolution in a polyacrylamide gel. For immunoprecipitation experiments, cells were lysed using glass beads in 2X HC lysis buffer (300 mM HEPES buffer pH 7.6, 100 mM KCl, 2 mM EDTA, 0.2% NP-40, and 0.2 mM DTT) containing protease inhibitor cocktail (11697498001, Roche) and 2 mM PMSF. Lysate cleared of cellular debris was incubated with antibody-conjugated beads for immunoprecipitation. Afterwards, beads were washed three times with 1X HC lysis buffer (150 mM HEPES pH 7.6, 250 mM KCl, 1 mM EDTA, 1 mM PMSF, 0.1% NP-40, and 1 tablet protease inhibitor cocktail per 100 mL volume) and twice with AC$_{200}$ wash buffer (20 mM HEPES pH 7.6, 1 mM EGTA, 200 mM KCl, 2 mM MgCl$_2$ 0.1% NP-40, 1 mM PMSF, and 1 tablet protease inhibitor cocktail per 100 mL volume). Protein elution was performed using 0.2 M glycine (pH 2). Eluted protein was precipitated by TCA precipitation and dissolved into SDS sample buffer prior to resolution in a polyacrylamide gel. Antibodies used at 1:1000 dilution were: Anti-FLAG (M2, Sigma), anti-GFP (7.1 and 13.1, Roche and gta20, Chromotek), and anti-Cdc2 (Y100.4, Santa Cruz). Ponceau S (Sigma) staining was used to visualize the total protein loaded.

**ChIP-qPCR and ChIP-seq**. ChIP experiments were performed as described[40]. 25 OD$_{595}$ of cells were harvested from YEA cultures grown to mid-log phase at 32 °C. Cells were fixed with 1% formaldehyde for 20 min at room temperature. For Erh1-ChIP, additional fixation with dimethyl adipimidate (Thermo Fisher Scientific) for 45 min at room temperature was performed. Cell pellets were suspended into 400 μL of ChIP lysis buffer (50 mM HEPES pH 7.5, 140 mM NaCl, 1 mM EDTA, 1% Triton, and 0.1% deoxycholate) and glass beads. Cell lysis was performed using a bead-beater and genomic DNA was sonicated using a Bioruptor (Diagenode) 12-cycles on medium power setting (30 s on, 30 s off) at 4 °C. Cellular debris was removed by centrifugation at 1500×g for 5 minutes at 4 °C. Lysate supernatant and brought up to 1 mL volume using ChIP lysis buffer and was precleared using 20 μL of Protein A/G-plus agarose slurry (Santa Cruz) with rotation for 1 h at 4 °C. Precleared lysates were centrifuged at 1000×g for 1 mine and lysate supernatant was transferred to a new tube for subsequent immunoprecipitation. Fifty microlitre of lysate was reserved as whole-cell extract input control. Anti-H3K9me2 (2μg per ChIP, ab115159, Abcam) and anti-GFP (10μg per ChIP, ab290, Abcam) antibodies were used for immunoprecipitation overnight at 4 °C. Antibodies were recovered using 20 μL of Protein A/G-plus agarose slurry (Santa Cruz) with rotation for 4 h at 4 °C. Beads were washed twice with ChIP lysis buffer, twice with ChIP high-salt buffer (50 mM HEPES pH 7.5, 500 NaCl, 1 mM EDTA, 1% Triton, 0.1% deoxycholate), twice with ChIP wash buffer III (10 mM Tris-HCl pH 8, 0.25 M LiCl, 0.5% NP-40, 0.5% deoxycholate, and 1 mM EDTA), and once with 1X TE pH 8. DNA was eluted by heating beads at 65 °C for 1 h in 100 μL of ChIP elution buffer

(1X TE pH 8, 1% SDS). After elution, NaCl was added to bring the concentration up to 100 mM. Crosslinking was reversed by heating at 65 °C overnight. RNA and protein were removed by treatment with RNase A (Sigma) and proteinase K (Thermo Fisher Scientific). DNA was purified using PCR purification kit (Qiagen) according to manufacturer instructions. Immunoprecipitated DNA or input DNA was analyzed by qPCR or Illumina sequencing.

For ChIP-qPCR analyses, experiments were performed using iTaq Universal SYBR Green Supermix (Biorad). Oligonucleotides used for ChIP-qPCR are listed in Supplementary Table 4. For ChIP-seq analyses, sequencing libraries were generated using NEBNext Ultra II DNA library prep kit for Illumina (NEB) according to the manufacturer's protocol. Samples were multiplexed and single-end reads were sequenced on the Illumina NextSeq 500 platform. Adapter trimmed reads were aligned to the *S. pombe* v2.29 reference genome using BWA-MEM. Correction for GC-content bias and input normalization was performed using the Deeptools suite[57] functions correctGCbias and bamCompare. Plots were generated using the plotProfile function.

**RNA-seq and RT-PCR**. Total RNA was prepared by harvesting 2 OD$_{595}$ of mid-log phase cells cultured in YEA medium at 32 °C followed by flash-freezing in liquid nitrogen. RNA was isolated using the hot-phenol method in equal volumes of AES buffer (50 mM sodium acetate pH 5.3, 10 mM DTA, 1% SDS) and acid-phenol. The mixture was incubated at 65 °C and vortexed every minute for 5 minutes. Afterwards, the slurry was transferred to Maxtract High Density tubes (Qiagen) and RNA was purified by chloroform extraction. Purified RNA was precipitated using Glycoblue (Thermo Fisher Scientific), sodium acetate pH 5.3 and isopropanol. Total RNA was treated with DNase I (Thermo Fisher Scientific) before subsequent analyses. For RNA-seq, ribosomal RNAs were depleted using the Ribo-Zero Gold rRNA Removal Magnetic Kit (Yeast) (Epicentre) prior to library construction using the NEBNext Ultra II Directional RNA library prep kit for Illumina (NEB) according to the manufacturer's instructions. Single-end sequencing was performed on the Illumina NextSeq 500 platform. Adapter trimmed reads were aligned to the *S. pombe* v2.29 reference genome using Tophat2[58]. Minimum and maximum intron sizes were set to 30 and 817, respectively. Normalization by reads per kilobase per million (RPKM) and coverage plots were performed using the Deeptools suite function[57] bamCoverage and plotProfile. For differential analyses, FPKM values were determined for each of the 7019 genes using Cufflinks[59]. Upregulated genes were defined as those with FPKM ratios ≥2 relative to WT. Heatmaps were generated using Java Treeview platform[60].

For RT-PCR experiments, cDNA was synthesized using Superscript III First-Strand Synthesis SuperMix (Thermo Fisher Scientific) using oligo-dT. Subsequent qPCR analyses were performed using iTaq Universal SYBR Green Supermix (Biorad). Oligonucleotides used for RT-PCR are listed in Supplementary Table 4.

**smFISH**. Single molecule RNA Fluorescence In-Situ Hybridization (smFISH) was performed with modifications to the manufacturer's protocol (Biosearch Technologies) and as described[61]. Mid-log cells were fixed in 3.7% formaldehyde and treated with Zymolyase 100 T for partial cell wall digestion. The cells were permeabilized in 70% ethanol and incubated overnight with probes against *ssm4* mRNA. CAL Fluor Red 590 labeled *ssm4* probes were designed using Stellaris Probe Designer tool (Supplementary Table 4) and synthesized by Biosearch Technologies. Stellaris RNA FISH hybridization and wash buffers were obtained from Biosearch Technologies. Cells were mounted in ProLong Gold antifade reagent with DAPI (Life Technologies) and imaged using a DeltaVision Elite fluorescence microscope (Applied Precision, GE Healthcare) with Olympus 100×/ 1.40 objective. Optical Z sections were acquired (0.2 µ step size, 20 sections) for each field. Images were deconvolved and all Z-stacks were projected into a single plane as maximum-intensity projections. Fiji (ImageJ) software was used for analysis. Cell boundaries were marked manually and the distribution of *ssm4* mRNA spots was calculated manually.

**Northern blot analysis**. Northern blot analysis was performed using total RNA isolated as described above for RNA-seq analyses. Ten microgram of total RNA was loaded per lane in a 1% formaldehyde agarose gel. T7 MAXIscript kit (Ambion) was used to generate α-P$^{32}$-UTP (PerkinElmer) labeled RNA probes and hybridizations were carried out using the NorthernMax kit (Ambion) according to manufacturer instructions. Uncropped gel images are provided as a Source Data file.

**Spotting assay**. Mid-log phase yeast cells were serially diluted fourfold and spotted onto YEA agar medium plates. Spotted plates were incubated at 18, 32, or 37 °C for 2–6 days. *erh1Δ* served as a known hypersensitive control at 18 °C[14], while *mmi1-ts6* served as a known hypersensitive control at 37 °C.

**Reporting summary**. Further information on experimental design is available in the Nature Research Reporting Summary linked to this article.

## Data availability

The atomic coordinates and structure factors for the EMC complex have been deposited to the Protein Data Bank (PDB) under the accession code PDB 6AKJ.

Genomic datasets are deposited in the Gene Expression Omnibus with accession numbers GSE119604 and GSE119605. All other materials are available from the corresponding authors upon reasonable request. Uncropped gel images and other source data underlying Figs. 1b–d, 3d, 4a, b and e, 6a, 8a, c, d are provided as a Source Data file. A Reporting Summary for this article is available as a Supplementary Information file.

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

## Acknowledgements

We thank Masayuki Yamamoto (National Institute of Basic Biology, Japan) for providing *mmi1-ts6* mutant strain, Diego Folco and Martin Zofall for their help with imaging and genomics methods, Xiaoling Bao, Rongsheng Ma, Xiaodan Liu and Tiantian Liu for help with biochemical assays, David Wheeler for help with bioinformatic analyses, Jemima Barrowman for editing the manuscript, and members of our labs for discussions. We also thank the staff of BL18U1 beamline at National Center for Protein Science Shanghai and Shanghai Synchrotron Radiation Facility for assistance during data collection. This work was supported by grants from the Ministry of science and technology of China (2016YFA0500700), the Strategic Priority Research Program of the Chinese Academy of Sciences (XDB08010101 and XDPB10), the Chinese National Natural Science Foundation (grant 31330018, 31500590, 31600600, and 31870760), the Fundamental Research Funds for the Central Universities (WK2070000095), the Postdoctoral Research Associate fellowship from the National Institute of General Medical Sciences (1Fi2GM123947-01), and by the Intramural Research Program of the National Institutes of Health, National Cancer Institute. This study used the Helix Systems and Biowulf Linux cluster at the National Institutes of Health.

## Author contributions

F.L., T.V.V., Y.S., and S.I.S.G. designed the study. G.X. crystallized the EMC. G.X., Y.J., and M.L. collected X-ray data. F.L. determined the crystal structure. G.X., B.Z., Z.X., and C.W. performed in vitro assays. T.V.V. performed majority of in vivo experiments including yeast strain constructions as well as genetic and genomic analyses. V.B. performed biochemical experiments; S.H. performed smFISH experiments; G.T. performed Northern blot experiments. S.I.S.G wrote the paper with input from T.V.V., F.L. and other authors.
