## [Peer Review File · Nature Communications]

Reviewers' comments:

Reviewer #1 (Remarks to the Author):

In this manuscript Xie et al. present their structure-function analysis of the Erh1-Mmi complex (EMC). They identified the minimal Erh1-interacting domain of Mmi1, characterized the interaction with biophysical methods and solved the structure by X-ray crystallography. The structural analysis revealed how Mmi1 binds the highly conserved dimer of Erh1, and identified Mmi1 W112 as a critical residue for complex formation. The authors employed the W112A mutant to determine the specific functional role of the EMC complex in gene regulation. The results show that the interaction is critical for heterochromatin formation and gene silencing at heterochromatic islands, and that Mmi1's role in transcription termination is independent of its interaction with Erh1.

This manuscript makes in-depth contributions to our understanding of a conserved post-transcriptional gene silencing mechanism as it reveals structure and function of a key interface between the sequence-specific adapter Mmi1 and Erh1, a highly conserved factor linked to RNA metabolism. This work builds on previous work by the authors and advances the mechanistic understanding of this silencing processes significantly. Provided the critical lack of evidence regarding Erh1 recruitment is addressed and provided the figures are improved as detailed below this manuscript is suitable for publication in Nature Communications.

Major points:

1. The author postulate that the Mmi1-Erh1 interface recruits Erh1 to heterochromatin islands. While the immunofluorescence localization data shows that Erh1 is lost from nuclear foci in the MmiW112A mutant it does not address the status of Erh1 mutant recruitment to heterochromatin islands. CHIP data similar to data previously published by the authors in Sugiyama et al. 2016 is necessary to establish evidence for the recruitment model.

Minor points:

2. Fig. 1 d-b: The legend should describe in more detail what is shown. Is it resin, elution?
3. Fig. 2 b: Legend should describe what the color scheme of the surface represents.
4. Fig. 2 c: The details in this panel are too small. This panel needs to be improved such that it to show the dimerization interface clearly.

5. Fig. 3 c: Not clear what the different sub-figures represent. Each should have its own legend or a clear label.

6. Fig. 4 c: The legends mentions mmi-ts3, but the figure is labeled with mmi-ts6. One of them is probably a typo and needs correction.

Reviewer #2 (Remarks to the Author):

This manuscript performs a structural and genetic dissection of the interaction between Mmi1 and Erh1 in fission yeast. Enhancer of Rudimentary is a conserved factor with unclear roles, so any new mechanistic information is interesting. Its function in silencing of meiotic genes in fission yeast may illuminate its conserved gene regulatory roles, as well as the general control of meiotic gene expression, which as the authors point out is sometimes altered in cancer. This paper reveals a conserved interaction interface in Erh1, which binds to a domain in Mmi1. Furthermore, using the structural and biochemical information, they create a separation of function allele of Mmi1 that affects the formation of the EMC complex but not other functions of Mmi1. Using this allele the authors then characterize the Erh1 dependent and independent functions on Mmi1 showing that a previously described role in non-canonical mRNA termination is independent of Erh1. These elegant experiments provide mechanistic insights into the separable functions of Erh1 and Mmi1. The conclusions are novel and well supported by the results, and will be of interest to a wide readership. I only have a few minor comments, mostly concerning supplemental data.

Minor comments:

Page 9, last paragraph. The authors should define what constitutes an Erh1-independent island. I assume this comes from Sugiyama et al 2016, but the reference is not cited in this paragraph. A supplementary table with the list of islands or genes used in figure S4A and B would help, as would a brief description (in materials and methods, or the figure legend) of the criteria used to classify the islands, and the quantity used in the correlation in figure 4B (average H3K9me over the island? Maximum?)

Supplemental figure 4 A: The legend indicating the color code of the aggregate profiles is missing.

Supplemental figure 4 B: I think that one of the axis is mislabeled, WT and mmi1W112A are flipped. Also, the diagonal (once the axes have been corrected) is uninformative and should be blanked out.

Supplementary figure 4 D: Please indicate the locus used for normalization to calculate “Fold enrichment” (leu1?), either here or in materials and methods.

Supplementary table 1: The text describes ITC experiments to test binding between Mmi1 95-122, but this table refers to Mmi1 WT. Change it for consistency.

Reviewer #3 (Remarks to the Author):

This clearly written manuscript reports structural and functional characterizations of the Erh1-Mmi1 complex (EMC) in *S. pombe*, in which the EMC plays important roles in the formation of facultative heterochromatin and silencing of meiotic genes. Through in vitro biochemical and biophysical analyses, the authors identified that an ~25 residue N-terminal segment of Mmi1 is responsible for binding Erh1, while the C-terminal RNA-binding YTH domain is dispensable for complex formation. They crystallized and solved the structure of the Mmi1 fragment in complex with Erh1, and found that a Erh1 dimer binds two Mmi1 molecules. The structure shows that the Erh1 dimer is assembled through evolutionarily conserved residues, and a tryptophan, W112, of Mmi1 is critical for interaction with Erh1. The structural data are nicely corroborated by in vivo studies, and a W112A mutant allele was used to dissect the functions the EMC and independent functions of Mmi1. They found that heterochromatin formation and gene silencing functions require the two proteins to stay together as a complex, while Mmi1 alone can fulfill the function at 3'-end formation of meiotic mRNAs and lncRNAs.

Altogether, this is a well-designed study with high quality data. The interpretation of the data is also well justified. I recommend acceptance of the manuscript essentially as is.

Minor question:

In Supplemental Table 2, the RMSD of bond angles is listed as 0.384 degree, which is unusually small. Please double check.

Response to reviewers' comments:

We are most grateful for the valuable feedback provided by the reviewers. In this revised version, we include new data to strengthen the main conclusions of our paper and address all of the concerns raised by the reviewers. We believe that our study is considerably improved and meets the standards of quality and novelty expected for publication in *Nature Communications*.

Some of the key changes that we have made are as follows:

- (1) As noted by the reviewers, our work reveals the structure and function of the nuclear RNA processing complex EMC (Erh1-Mmi1 complex). Importantly, Erh1 interacts with Mmi1 through a highly conserved dimer interface and this interaction is critical for the assembly of facultative heterochromatin islands and silencing of gametogenic genes. To further strengthen our conclusions, reviewer #1 suggested investigating the localization of Erh1 at heterochromatin islands in *mmi1^{W112A}* mutant cells by chromatin immunoprecipitation (ChIP). We now include the results of ChIP analyses showing that Mmi1-Erh1 interaction is required for the recruitment of Erh1 to heterochromatin islands (see Fig. 4f). This result underscores the crucial role of EMC assembly in facultative heterochromatin formation and gene silencing.
- (2) Tethering of gametogenic transcripts to the nuclear foci is believed to be important for preventing their translation and expression in mitotic cells. We now show that EMC assembly, rather than the individual Mmi1 or Erh1 protein, is required for nuclear sequestration of gametogenic transcripts by utilizing the *mmi1^{W112A}* mutant that specifically disrupts interaction between Mmi1 and Erh1 proteins without affecting their cellular levels. The results presented show that unlike wild-type cells, *mmi1^{W112A}* mutant cells show export of gametogenic gene transcripts from the nucleus into the cytoplasm, suggesting that EMC assembly is critical for nuclear retention of target gene transcripts.
- (3) Our analyses using separation of function *mmi1^{W112A}* mutant has revealed that Mmi1 participates in two overlapping mechanisms to control expression of genes located adjacent to cis-acting long non-coding RNAs (lncRNAs). For example, while Mmi1 represses *pho1* gene as part of EMC, it also mediates transcription termination of an upstream lncRNA, to prevent it from invading and repressing *pho1*. We find that strong de-repression of *pho1* observed in *mmi1^{W112A}* mutant cells is obscured in *mmi1Δ* cells presumably due to the lncRNA transcribing into *pho1* as a consequence of its failed termination. However, it remained unclear whether indeed lncRNA is responsible for repression of *pho1* in *mmi1Δ* cells. We now show direct evidence that deletion of lncRNA abolishes the *pho1* repression observed in *mmi1Δ*, as compared to *mmi1^{W112A}* cells (see Fig. 8c). This finding further supports our conclusions that in addition to its role as part of EMC, Mmi1 regulates gene expression via a mechanism involving termination of regulatory lncRNAs.

- (4) We have paid special attention to the style and formatting guidelines. We have highlighted the additions we have made to the text (red font color) and include a Source Data file that includes uncropped versions of all gels and blots presented in the main and supplementary figures. The source data is mentioned in all relevant figure legends.

Below are our responses to the specific comments of the reviewers, which are indicated in italics:

Reviewer #1:

We are thankful to the reviewer for providing helpful suggestions. The reviewer commented, *“This manuscript makes in-depth contributions to our understanding of a conserved post-transcriptional gene silencing mechanism as it reveals structure and function of a key interface between the sequence-specific adapter Mmi1 and Erh1, a highly conserved factor linked to RNA metabolism. This work builds on previous work by the authors and advances the mechanistic understanding of this silencing processes significantly.”* As mentioned above, we now provide additional analyses to address reviewer’s suggestions.

Specific comments have been addressed as follows:

1. *The author postulate that the Mmi1-Erh1 interface recruits Erh1 to heterochromatin islands. While the immunofluorescence localization data shows that Erh1 is lost from nuclear foci in the MmiW112A mutant it does not address the status of Erh1 mutant recruitment to heterochromatin islands. ChIP data similar to data previously published by the authors in Sugiyama et al. 2016 is necessary to establish evidence for the recruitment model.*

As recommended, we have added results of ChIP analyses showing that *mmi1^{W112A}* cells are defective in recruitment of Erh1 to heterochromatin islands. The inclusion of this data further strengthens the conclusions presented.

2. *Fig. 1 d-b: The legend should describe in more detail what is shown. Is it resin, elution?*

We added a short description of the GST-pull down assay. The figures show the protein complex that is pulled down by GST resin.

3. *Fig. 2 b: Legend should describe what the color scheme of the surface represents.*

We added the color scheme description. Also, we have included a color bar in Fig. 2b which shows the color scheme more clearly.

4. *Fig. 2 c: The details in this panel are too small. This panel needs to be improved such that it to show the dimerization interface clearly.*

We improved the figure by presenting an enlarged dimer interface panel that shows the interaction details more clearly.

5. Fig. 3 c: Not clear what the different sub-figures represent. Each should have its own legend or a clear label.

As recommended, we have included a brief description for each panel.

6. Fig. 4 c: The legends mentions *mmi-ts3*, but the figure is labeled with *mmi-ts6*. One of them is probably a typo and needs correction.

We are thankful to the reviewer for pointing out this error. The strain used is *mmi1-ts6*. We have modified the figure legend to correct this error.

Reviewer #2:

The reviewer commented, “*These elegant experiments provide mechanistic insights into the separable functions of Erh1 and Mmi1. The conclusions are novel and well supported by the results, and will be of interest to a wide readership. I only have a few minor comments, mostly concerning supplemental data*”. We are grateful to the reviewer for supporting publication of our work and for helpful suggestions.

Specific comments have been addressed as follows:

Page 9, last paragraph. The authors should define what constitutes an Erh1-independent island. I assume this comes from Sugiyama et al 2016, but the reference is not cited in this paragraph. A supplementary table with the list of islands or genes used in figure S4A and B would help, as would a brief description (in materials and methods, or the figure legend) of the criteria used to classify the islands, and the quantity used in the correlation in figure 4B (average H3K9me over the island? Maximum?)

As suggested, we have included a supplementary table listing EMC-dependent and EMC-independent heterochromatin islands, along with the criteria used to classify the islands (see Supplementary Table S3). Heterochromatin islands that show depleted H3K9me2 levels in *mmi1^{W112A}* and *erh1Δ*, as compared to wild-type cells, are categorized as EMC-dependent, while others that showed no major changes in H3K9me levels are referred to as EMC-independent heterochromatin islands. Also, the average H3K9me2 enrichment is used in the correlation computation. This is now mentioned in the Supplementary Fig. 4 legend.

Supplemental figure 4 A: The legend indicating the color code of the aggregate profiles is missing.

We are thankful to the reviewer for pointing out this oversight. We have now added color codes for both panels in Supplemental Figure 4a.

Supplemental figure 4 B: I think that one of the axis is mislabeled, WT and mmi1W112A are flipped. Also, the diagonal (once the axes have been corrected) is uninformative and should be blanked out.

We have modified the figure and corrected the errors pointed out by the reviewer.

Supplementary figure 4 D: Please indicate the locus used for normalization to calculate “Fold enrichment” (leu1?), either here or in materials and methods.

The modified figure legend now indicates the locus (*leu1*) used for normalization. We have also made a similar correction to Supplementary Fig. 5a.

Supplementary table 1: The text describes ITC experiments to test binding between Mmi1 95-122, but this table refers to Mmi1 WT. Change it for consistency.

We have modified the table to correct this error.

Reviewer #3:

The reviewer commented, “*Altogether, this is a well-designed study with high quality data. The interpretation of the data is also well justified. I recommend acceptance of the manuscript essentially as is.*” We are thankful to the reviewer for supporting publication of our work. The minor question by the reviewer has been addressed as follows:

In Supplemental Table 2, the RMSD of bond angles is listed as 0.384 degree, which is unusually small. Please double check.

As suggested, we have checked the crystallography parameters carefully. The fact that the RMSD of bond angles seems unusually small is because we used the “optimize the geometry and/or B-factor restraint weights” function in the final round of refinement using Phenix.refine. This results in a significantly better refinement and prevents overfitting as has been described in the document Phenix.refine (<https://www.phenix-online.org/documentation/faqs/refine.html#optimization-methods>). The document also mentions that after the weight optimization, the RMSD angles and bonds are usually small but reasonable.

REVIEWERS' COMMENTS:

Reviewer #1 (Remarks to the Author):

The authors have addressed all my concerns and I fully support the publication of their manuscript in Nature Communications.